# Mass agnostic jet taggers

**Layne Bradshaw[1]⋆, Rashmish K. Mishra[2], Andrea Mitridate[2] and Bryan Ostdiek[1]**

**1** Institute of Theoretical Science and Center for High Energy Physics,
Department of Physics, University of Oregon, Eugene, Oregon 97403, USA
**2** INFN, Pisa, Italy and Scuola Normale Superiore, Piazza dei Cavalieri 7, 56126 Pisa, Italy

⋆ layneb@uoregon.edu

## Abstract

Searching for new physics in large data sets needs a balance between two competing effects—signal identification vs background distortion. In this work, we perform a systematic study of both single variable and multivariate jet tagging methods that aim for this balance. The methods preserve the shape of the background distribution by either augmenting the training procedure or the data itself. Multiple quantitative metrics to compare the methods are considered, for tagging 2-, 3-, or 4-prong jets from the QCD background. This is the first study to show that the data augmentation techniques of Planing and PCA based scaling deliver similar performance as the augmented training techniques of Adversarial NN and uBoost, but are both easier to implement and computationally cheaper.



# 1   Introduction

As the search for new resonances continues at the Large Hadron Collider (LHC), it is increasingly important to develop and apply search strategies that are sensitive to a wide class of signals. For hadronically decaying resonances, there has been considerable effort in the past to develop various methods, targeted at the boosted regime ($p_T \gg m$) of these resonances. Such boosted resonances appear in many generic Beyond Standard Model (BSM) scenarios, as well as in hadronic channels of boosted W/Z in the Standard Model (SM) itself. In the boosted regime, the resulting jets from the hadronic decay of these resonances are merged, and the result is a fat jet of wide radius. Using the difference in radiation pattern inside these fat jets, captured by various substructure variables, single variable (SV) [1,2] as well as multi-variable (MV) machine learning based methods [3–24] have been shown to allow a good discrimination of these signals from QCD background (see [25, 26] for a review of machine learning based techniques in high energy physics).

While entirely focusing on the best discriminant to distinguish between signal and background is desirable, it is only a first step. In realistic searches for these resonances, one needs to model the background with confidence, given that QCD is hard to estimate entirely analytically. This is usually accomplished by looking at distributions of variables in which the background is smooth and featureless, while the signal is not—an example of such a variable being the invariant mass of the jet. Using sideband analysis or control regions, one can model the background, and therefore look for new resonances using a bump hunt strategy.

The substructure of a fat jet is related to kinematic variables such as the jet mass, $m$, and transverse momentum, $p_T$. As a result, the application of any classifier for signal isolation tends to distort the background distribution for $m$ and $p_T$. This leads to introducing spurious features in the distributions, making a bump hunt harder to implement with statistical confidence. It is not surprising that such a distortion for the background distribution occurs, because a good discriminant should reject a large fraction of background events, so that the events that survive are necessarily signal like, and hence the background distribution starts to look signal like. The right optimization requires taking these two competing effects into account—a strong signal discrimination vs an undistorted background distribution.

Specifically, there are two side effects that come as a result of the correlation of the jet mass with the classifier output. The first is that the classifier is only good for a signal of a given mass. This is less than ideal as a broad search strategy for new physics. One would either need to train multiple classifiers to cover the mass range, or need to use other techniques such as parametrized networks [5, 27]. The other side effect is related to systematics. If the only

background that makes it through the selection criteria looks exactly like the signal, it can be hard to estimate the level of background contamination. While unintuitive, it can be better to have a classifier which removes less background, if it does so in such a way that the systematics are decreased. The overall goal is to maximize the significance, which is approximately given by $S/\sqrt{B + \sigma_{\text{sys}}^2}$. Allowing more background can lead to a better significance if it decreases the systematic uncertainty $\sigma_{\text{sys}}$.

Recent work, based on both single variable and multivariate approaches have addressed this constrained optimization problem. For example, a decorrelated $\tau_{21}$, called $\tau_{21}^{\text{DDT}}$ has been shown to be effective in keeping background distributions unaffected [28, 29]. While this single variable method has the advantage of being simpler to implement, it will not be useful for more complicated boosted jets. Multivariate methods, while more powerful in general as compared to single variable based methods, are also prone to distorting the background distributions more, and require more sophisticated *training augmentation* based approaches. For example, multivariate methods based on Boosted Decision Trees (BDT) use a modified algorithm called uBoost to perform this constrained optimization [30]. Multivariate methods based on Neural Networks (NN) use an adverserial architecture [27, 31, 32] to accomplish the same. However, these multivariable methods are significantly more involved and require tuning additional hyperparameters for optimal performance. In a recent work [33], the ATLAS collaboration has studied mass decorrelation in hadronic 2-body decays for both single and multivariate approaches.

In addition to these, there are *data augmentation* based approaches that aim for a middle ground.[1] The idea is to decorrelate the input to multivariate methods, so that any dependence on a given background variable is reduced significantly. While these methods are not as efficient in keeping the distributions undistorted, they are quick to implement and still enjoy the power of multivariate discrimination. Two such approaches, PCA [7, 28] (based on principal component analysis, from which it derives the name) and Planing [4, 9] are shown to be efficient in benchmark cases.

There is a general need to compare and understand the advantages and limitations of these methods, when requiring both high signal isolation and undistorted background distribution. A classification of these methods, and quantifying their performance using suitable metrics, for varying levels of signal complexity (in terms of prongedness) is desired. Depending on the situation at hand, one may want to work with higher/lower signal efficiency or lower/higher background rejection, for a given background distortion. This should be quantified for various methods and signal topologies. This can give a clear picture of when is a given method suitable, and how to augment one with the other if needed. This is the aim of the present work.

The outline of the paper is as follows. A brief overview of the Monte Carlo simulation used to generate the signal and background events is given in Sec. 2. In Sec. 3, we classify and describe the representative methods for decorrelation, first focusing on the general idea and then on specific details. We present the results in Sec. 4, comment on future work in Sec. 5, and conclude in Sec. 6. Appendix A shows the results of the parameter sweep used to choose the adversarial network studied in this work. A comparison of popular histogram distance metrics is shown in App. B. A side-by-side graphical comparison of all of the decorrelation methods applied to all of the signals considered is shown in App. C. Code to reproduce our results can be found on GitHub.

---

[1]In the machine learning literature, data augmentation is a technique to modify an input and add to the existing training set. This can make a classifier more robust to noise or underlying symmetries. We use data augmentation instead to remove information that we don't want to be learned.

## 2   Simulation details

In this section, we provide the details about the Monte Carlo simulated dataset used in this study. While this study does not rely on any specific model where the fat jets with some level of prongedness come from, we choose to work with a model which can give signals with 2- as well as 3- and 4-pronged jets, in suitable parts of parameter space. Studying higher pronged signals is useful in the context of mass decorrelation methods—apart from broadening the scope of the study, higher pronged jets are also sufficiently distinct from the background QCD jets, so that the importance of de-sculpting of mass distribution is changed compared to lower pronged jets. We quantify these statements in the next sections.

The model considered is based on warped extra-dimensional RS models with more than 2 branes (see Ref. [34] for theory and [35–37] for phenomenological details). The relevant degrees of freedom for our case are the KK modes of the EW gauge boson (massive spin-1 EW charged particles, denoted by $Z_{KK}/W_{KK}$) and the radion (a massive spin-0 singlet under SM, denoted by $R$). In this "extended" RS model, the radion coupling to tops/higgs/gluons is highly suppressed as compared to usual RS models, so that the dominant way to produce the radion is through the spin-1 KK EW gauge boson's decay into SM gauge bosons and a radion. This further leads to the dominant decay modes of the radion to be into SM W/Z. In the fully hadronic decay channel of W/Z from radion decay, one expects 4-pronged jets when the radion and/or the intermediate W/Z are boosted (see Ref. [37] for a detailed discussion on various regimes of boosted topology depending on the mass of the radion). The spin-1 KK EW gauge boson couples to SM particles like its SM counterpart. For preparing a 2-pronged signal sample, we use the process $p + p \rightarrow Z_{KK} + j$, $Z_{KK} \rightarrow jj$, for a 200 GeV mass $Z_{KK}$. The produced $Z_{KK}$ is boosted due to recoil with the first jet, so that in its fully hadronic decay, we get a 2-pronged jet. For a 3-pronged jet, we use the process $p + p \rightarrow Z_{KK} \rightarrow t\bar{t}$, with the usual 3-pronged fully hadronic top decay. In this case, the $Z_{KK}$ is not boosted. Choosing the mass of $Z_{KK}$ to be 1500 GeV, the tops from its decay are sufficiently boosted, so that we get a boosted 3-pronged sample. Finally, for the 4-pronged case, we consider $p + p \rightarrow Z_{KK} \rightarrow Z(\rightarrow \nu\bar{\nu})R(\rightarrow WW \rightarrow jjjj)$. The $Z_{KK}$ mass is taken to be 1500 GeV, and is produced unboosted. For a light radion of mass 200 GeV, the radion is produced boosted, and in its fully hadronic decay mode through Ws, we get a 4-pronged jet. Note that if one of the W from the radion decays leptonically, we would get non-isolated leptons inside a 2-pronged jet, which would be rejected by usual isolation criteria. Further, in the case of radion decay to two Zs, if one of the Z decays invisibly, we would again be led to a 2-pronged jet. We avoid these complications by simply focusing on the fully hadronic decay mode of the radion through Ws.

The details of the signal process considered are shown in Tab. 1, along with the masses and the kinematic cuts chosen (at generation level) to produce boosted jets of desired prongedness. The background for these signals is taken to be QCD jet, generated by $p + p \rightarrow Z + j$, $Z \rightarrow \nu\bar{\nu}$, at leading order in QCD coupling. A sample size of 500K is generated for each signal category, while 1M events are generated for the QCD background,[2] using MADGRAPH@AMC 2.6.4 [38] for parton-level events generation (14 TeV center of mass energy), PYTHIA 8 [39] for parton showers and hadronization, and DELPHES 3.4.1 [40] for detector simulation. Jets are constructed from the track and tower hits, using the anti-$k_t$ algorithm implementation in FAST-JET, with a jet radius $R = 1.2$. The clustered jets are required to satisfy $p_{T,J} > 500$ GeV and $-2.5 \leq \eta_J \leq 2.5$. A mass cut of $50 \leq m_J(\text{GeV}) \leq 400$ is further imposed on the groomed mass of the jet, where grooming is performed by Pruning [41] with Cambridge-Aachen algorithm, with $z_{cut} = 0.1$ and $R_{cut} = 0.5$. The highest $p_T$ jet is considered as the candidate jet, from which the higher level NN inputs are constructed using the NSUBJETTINESS module in FASTJET for axis choice of ONEPASS KT AXES, for the same jet radius used in the construction

---

[2]We do not use jet matching or merging and only take the hardest jet in the event.

Table 1: Details of the signal process used in the event generation, along with the choice of parameters and generation level kinematic cuts.

| Prong | Process | Parameters (TeV) | Kinematic Cuts (GeV) |
|---|---|---|---|
| 2P | $p + p \rightarrow j + Z_{KK}, \ Z_{KK} \rightarrow j\ j$ | $m_{KK} = 0.2$ | $p_{T,\min} = 50, \ p^{\geq 1}_{T,\min} = 400$ |
| 3P | $p + p \rightarrow Z_{KK} \rightarrow t\ \bar{t}$ | $m_{KK} = 1.5$ | $p_{T,\min} = 50$ |
| 4P | $p + p \rightarrow Z_{KK} \rightarrow Z(\nu\bar{\nu}) + R(jjjj)$ | $m_{KK} = 1.5, \ m_{R} = 0.2$ | $p_{T,\min} = 50$ |

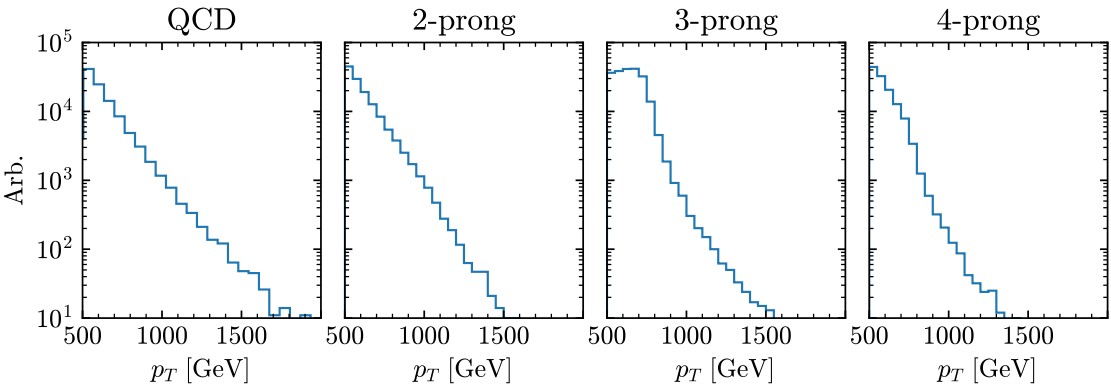

Figure 1: Distributions of the transverse momentum of the hardest jet.

of the original jet.

After the pre-selection cuts, the original 1M sample of QCD jets is cut down to 151 559. Similarly, the 500k events for the different BSM jets are reduced to 187 659, 303 917, and 177 418 for the 2-, 3-, and 4-prong signals, respectively. The $p_T$ distributions for the different samples are shown in Fig. 1. Training the machine learning algorithms is done on 70% of the combined datasets with 15% set aside for validation and 15% for independent testing.

## 3 Classification of Methods

In this section, we introduce various methods for decorrelating the mass distribution from classifier output. For classifiers, we consider single variable, such as $\tau_{21}$, as well as multivariate based architectures such as BDTs and NNs. We note that typically, multivariate analysis refer to shallow NNs or BDTs, as opposed to the more modern machine learning architectures. For mass decorrelation, we consider either augmenting the data, to reduce the correlation of jet mass from the input to the classifier, or augmenting the training, where the optimization procedure is modified to decorrelate the classifier output from mass. We also introduce the benchmark classifiers, which are needed for comparison.

### 3.1 Classification without decorrelation

To allow a comparison for the performance of various decorrelation methods, we need to introduce the corresponding benchmark methods, which do not take any decorrelation into account. Jet classification is often done using the substructure within the jet. The $N$-subjettiness

observables $\tau_N^{(\beta)}$ [1, 2, 42] can quantify the substructure, and are defined as

$$\tau_N^{(\beta)} = \frac{1}{p_{T_J}} \sum_{i \in \text{Jet}} p_{T_i} \min \left\{ \Delta R_{1i}^\beta, \ \Delta R_{2i}^\beta, \ \cdots, \ \Delta R_{Ni}^\beta \right\}, \tag{1}$$

where $p_{T_J}$ is the transverse momentum of the whole jet, $p_{T_i}$ is the transverse momentum of the $i^{\text{th}}$ constituent of the jet, $\Delta R_{Ai}$ is the distance between axis $A$ and constituent $i$ and $\beta$ is a real number. The distance is defined as

$$\Delta R_{Ai} = \sqrt{\Delta \phi_{Ai}^2 + \Delta \eta_{Ai}^2} \, . \tag{2}$$

Suitable choices of the sub-jet axes lead to small values for different $\tau_N^{(\beta)}$. For instance, a boosted, hadronically-decaying W will have two hard partons in the jet. If the axes are chosen to be along the directions of these two partons, the value of $\tau_2^{(\beta)}$ will be much lower than $\tau_1^{(\beta)}$ where only one axis is considered. In contrast, a QCD jet will have a radiation pattern taking up more of the jet area, leading to constituents further away from the axes; both $\tau_1^{(\beta)}$ and $\tau_2^{(\beta)}$ will be relatively large. With this, a common method for classifying jets with 2-prong structure is to examine the ratio between the two,

$$\tau_{21} \equiv \frac{\tau_2^{(1)}}{\tau_1^{(1)}}. \tag{3}$$

For our 2-prong signal (described in more detail in Sec. 2), using $\tau_{21}$ results in an area under the receiver operating characteristic curve (AUC) of 0.747. An AUC of 0.5 is the equivalent of randomly guessing, and an AUC of 1.0 is a perfect classifier. Thus, $\tau_{21}$ is a simple, single observable which significantly aids in discriminating 2-prong jets. When looking for boosted jets with more prongs, an analogous strategy is applied. For 3-prong jets, we use $\tau_{32} = \tau_3^{(1)}/\tau_2^{(1)}$ and the observable $\tau_{43} = \tau_4^{(1)}/\tau_3^{(1)}$ is used for 4-prong jets. The corresponding AUCs are 0.819 and 0.938. Once again, these simple single variable observables are strong discriminators of the corresponding signal topologies.

We use $\tau_{21}$, $\tau_{32}$, and $\tau_{43}$ as examples of single variable based classifiers. The benefit of these is that the variable is physics based and the systematics can be readily studied. However, a single variable may not be able to take advantage of the correlations of other observables in the data (e.g. see [43] for a BSM example). To fully incorporate all of the information, multivariate analysis is needed. We study two such multivariate methods, based on boosted decision tree and neural network architectures, which have been shown to lead to increased discrimination.[3]

The authors of [44] introduced a minimal but complete basis for a jet with $M$-body phase space. In particular, they showed that the dimension of the $M$ body phase space is $3M - 4$ and can be spanned using combinations of the $\tau_N^{(\beta)}$. In our study, we examine jets with up to 4-prong structure. We use a 5-body phase space for our multivariate analyses, as the performance is seen to saturate for 4-prong signals for a larger basis. For jets with fewer prongs, the 5-body basis is over-complete and the results saturate as well. This 5-body phase space basis is given as

$$X = \left\{ \tau_1^{(0.5)}, \tau_1^{(1)}, \tau_1^{(2)}, \tau_2^{(0.5)}, \tau_2^{(1)}, \tau_2^{(2)}, \tau_3^{(0.5)}, \tau_3^{(1)}, \tau_3^{(2)}, \tau_4^{(1)}, \tau_4^{(2)} \right\} \, . \tag{4}$$

These observables are used as the inputs for all of the multivariate approaches studied here. While this basis covers the substructure, the overall scale of the jet is not taken into account.

---

[3]We do not use convolutional neural networks (jet images), but only focus on the jet substructure variables to keep the data representation constant across all methods.

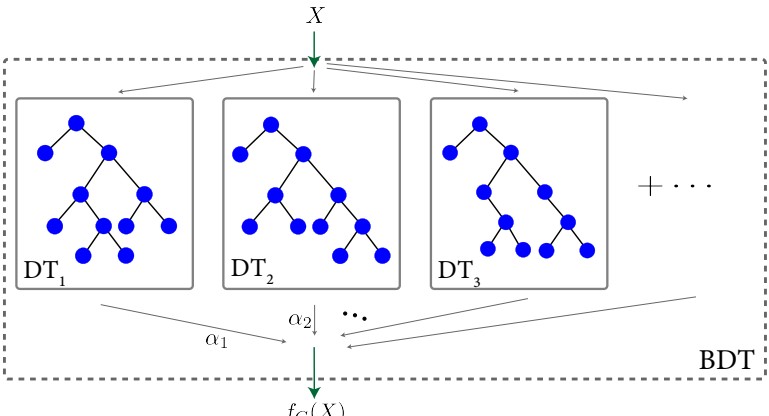

Figure 2: The architecture of a BDT. We take the BDT to be made of 150 DTs, with a max depth of 4. The input to the BDT are the variables that span the 5-body jet phase space, see Eq. (4). The indicated parameters $\alpha_i$ represent the weight associated with the particular DT.

Including the overall scale by using the transverse momentum or jet mass allows the classifiers to achieve better background rejection for a given signal efficiency, but at the expense of more sculpting. In the interest of not sculpting, and to have a fair comparison with the single variable taggers, we do not use the transverse momentum or the jet mass as an input for the machine learning algorithms.[4]

The first multivariate method we consider is based on a boosted decision tree (BDT) architecture. A BDT is made of decision trees (DT), which are a tree of binary decisions on various variables, leading to a final binary classification of data. Boosting is the technique to allow an ensemble of DT with weak predictions to build an overall strong classifier, thereby *boosting* the performance. The DTs are ordered such that each subsequent DT learns on the failures of its predecessors, by assigning higher weights to the misclassified events. Figure 2 shows the architecture of a BDT successively made from many DTs. BDTs have the advantage of being faster to train, less prone to overfitting and easier to see inside the box, as compared to methods based on Neural Networks (NN). However, they are more sensitive to noisy data and outliers.

Before training, the inputs are first scaled using the STANDARDSCALER of SCIKIT-LEARN so that each variable has zero mean and unit variance on the training set. The data is split into three separate sets, one for training, one for validation, and one for testing. The same STANDARDSCALER is used for all of the sets. We use the standard implementation of the gradient boosting classifier within the SCIKIT-LEARN framework [45]. In particular, we use 150 estimators, a max depth of 4, and a learning rate of 0.1. This leads to good discrimination, with an AUC of 0.863—a 15% increase compared to using just $\tau_{21}$—for the same two-prong jets as before.

The second multivariate method we consider is based on neural networks. Figure 3 shows the basic setup of our network, which is implemented in the KERAS [46] package with the TENSORFLOW backend [47]. Unless otherwise stated, all neural networks in this study use the same architecture, with three hidden layers of 50 nodes each. The nodes are activated using the Rectified Linear Unit (ReLu). The last layer contains a single node with a Sigmoid activation function so that the output is a number between 0 and 1. We experimented with

---

[4]The $\tau_1^{(2)}$ observable is related to the ratio of $m/p_T$, so including the transverse momentum would allow the multivariate analysis the possibility to learn the jet mass.

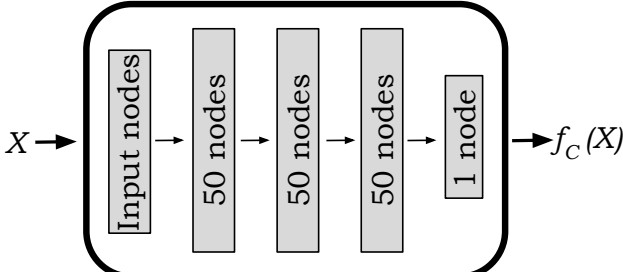

Figure 3: Many of the methods explored in this paper use a neural network classifier. For consistency, we always use a network with three hidden layers, each of which has 50 nodes and uses the ReLu activation function. The output is a single node with a sigmoid activation function. Our input data are the 11 $\tau_N^{(\beta)}$ variables of 5-body jet phase space shown in Eq. (4).

increasing or decreasing the number of layers, and found that three hidden layers is where performance saturated. Adding more nodes was not found to be helpful.

Training is done using the ADAM optimizer [48] to minimize the binary cross entropy loss function, which is given by:

$$L_{\text{classifier}} = -\frac{1}{N}\sum_i^N w_i \left[ y_i \ln f_C(X_i) + (1-y_i)\ln\left(1 - f_C(X_i)\right)\right], \qquad (5)$$

where $y_i$ is the true label, $f_C(X_i)$ is the network output, and $w_i$ is the weight for the $i^{\text{th}}$ event. It is standard for all of $w_i$ to be taken to be one, but in the case of unbalanced classes with significant difference in the number of training samples, it is useful to set $w_i$ to a specific value per class so that the effective number of training samples for each class becomes equal; these are called class weights. We implement class weights throughout as it was found to improve the classifiers, even though we do not have badly imbalanced classes. In Sec. 3.2.2, we explore another application of using weights during training.

The learning rate is initially set to $10^{-3}$. The loss is computed on the validation set after each epoch of training to ensure that the network is not over fitting. If the validation loss has not improved for 5 epochs, the learning rate is decreased by a factor of 10, with a minimum of $10^{-6}$. Training is stopped when the validation loss has not improved for 10 epochs. Training usually takes between 30-40 epochs.

To have a fair comparison with the BDT, the network is trained on the same training set, using the same pre-processing. In addition, a common test set is used for all comparisons. The depth of the network allows it to learn more of the non-linearities between the input features than the boosted decision tree, yielding a AUC of 0.872. This is only a 1% increase in the AUC, but this can have large impacts on the potential discovery of new physics. For instance, at a fixed signal efficiency of 0.5, the background rejection increases from a factor of 13 to a factor of 15, allowing for 16% more background rejection.[5]

A summary of the application of the three different methods presented so far is in Fig. 4. The left panel shows the ROC curves, where better classifiers are up and to the right. In what follows, we will always use a solid line to denote a neural network based classifier, a dashed line for a BDT, and a dotted line for a single variable analysis. The two multi-variate analysis are similar and do much better than the single variable $\tau_{21}$. The right panels highlight the main

---

[5]Here, and in the rest of the paper, we define the background rejection over the whole jet mass range considered: $50 \leq m_J(\text{GeV}) \leq 400$. We expect this choice to give the same qualitatively result which would be obtained by defining the background rejection on a smaller mass window (more details on this can be found in section 4.).

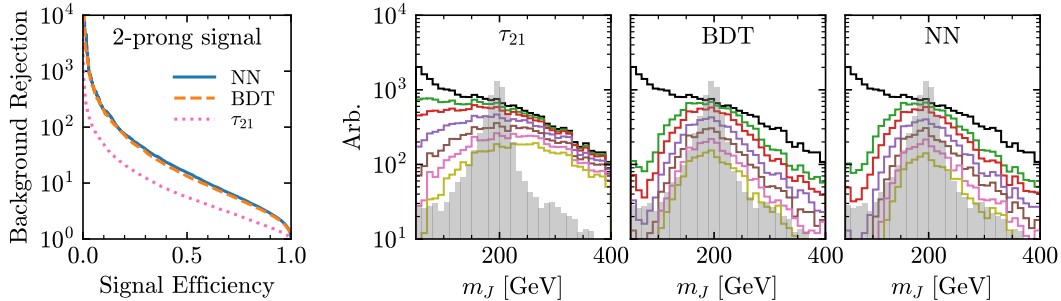

Figure 4: The left panel shows the ROC curves for three traditional methods, two based on machine learning, to classify a 2-prong signal jet from a QCD jet. The machine learning based methods achieve an area significantly higher than the single variable $\tau_{21}$ based classifier. The right panels show the *background only* distributions for successively tighter cuts in the solid lines: signal efficiency of 1.0 (black), 0.95 (green), 0.9 (red), 0.8 (purple), 0.7 (brown), 0.6 (pink) and 0.5 (yellow). The signal with no cuts is shown in the filled-in, grey distribution. The only background events which pass the cuts end up having masses similar to that of the signal, even though the machine learning models do not have access to the mass.

problem explored in this work. The solid black line and the grey, shaded regions show the jet mass distributions for the QCD background and the 2-prong signal, respectively. The different colored lines show the resulting QCD only distribution when cutting to signal efficiencies of 0.95, 0.9, 0.8, 0.7, 0.6, and 0.5. The $\tau_{21}$ classifier removes much of the QCD background at low jet masses, but allows many more events at high masses, so the background efficiency changes drastically as a function of the jet mass. This is even worse for the multivariate analyses, which drastically sculpt the background distributions. Even though they are only using substructure information, and do not have access to the overall scale of the jet, the QCD events that make it through are peaked at the signal mass. This better background rejection comes at the cost of having both the signal and background shapes becoming very similar, which makes estimating systematic uncertainties much harder.

With this motivation, we now turn to the different approaches of decorrelating the output of a classifier with a given variable such as jet mass. These approaches broadly fall into two categories. The first is to augment the data on which the model is trained, while leaving the training procedure unchanged. The second category is to not augment the data, but to alter the training algorithm itself. We discuss these two in turn next.

## 3.2 Decorrelation based on data augmentation

The general idea of data augmentation is to reduce as much as possible the correlation of the classifier input to the jet mass. This can be done for both single and multivariable methods. For single variable classifier, this can be done analytically, which we review below. For multivariable classifiers, the decorrelation must be done numerically, which we study using two recently proposed methods: *Planing* [9] and *PCA-based rescaling* [7,28]. Both of these methods can be used for NNs and BDTs; in this section we only show the examples for the NN. These methods are fast, and have little application-time computation cost.

### 3.2.1 Analytic decorrelation

For classification based on a single variable such as $\tau_{21}$, analytic decorrelation methods have been proposed [28,29], where a modified variable is constructed which is explicitly designed

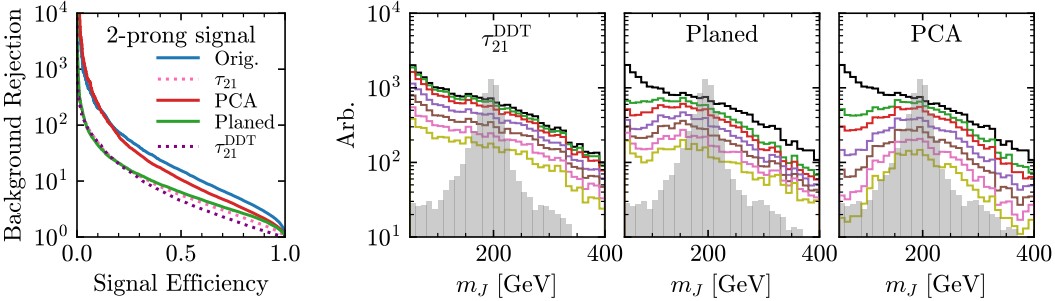

Figure 5: The left panel shows the ROC curves for the data augmented neural network methods of PCA and planing as well as the single variable DDT. The network trained on PCA-rescaled data is the best classifier, followed by the network trained on planed data. Both MV decorrelation techniques result in better classification than the single variable $\tau_{21}^{\text{DDT}}$ based classification. The right panels show the *background only* distributions for successively tighter thresholds for the DDT, Planed, and PCA classifiers: signal efficiency of 1.0 (black), 0.95 (green), 0.9 (red), 0.8 (purple), 0.7 (brown), 0.6 (pink) and 0.5 (yellow). For context, the 2-pronged signal distribution is shown as grey filled-in region. All three methods reduce the background sculpting when compared to their Fig. 4 counterparts. A full side-by-side comparison for 2, 3, and 4 prong signals is shown in App. C.

to preserve the background distribution. The appropriate scaling variable for QCD jets is the dimensionless ratio $\rho = \log(m^2/p_T^2)$. A plot of $\tau_{21}$ vs $\rho$ shows that background jets in different $p_T$ ranges are linearly shifted from one another, and that there is a linear relation between $\tau_{21}$ and $\rho$ for a certain range of $\rho$. With this information, the decorrelation with mass can be performed in two steps. The $p_T$ dependence is removed by defining $\rho' = \rho + \log(p_T/\mu)$ where the value of $\mu$ is chosen phenomenologically (taken to be 1 GeV in [28]). The linear correlation between $\tau_{21}$ and $\rho'$ can be removed by considering a modified variable—the so-called "Designed Decorrelated Tagger", $\tau_{21}^{\text{DDT}} = \tau_{21} - M\rho'$, where $M$ is the numerically calculated slope of the $\tau_{21}$ vs $\rho'$ curve. Apart from being simple to implement, the background systematics are easier to study because the method only involves a linear shift of the original observable. However, this method fails to generalize to more complex topologies, as there is not a simple linear relation between $\tau_N^{(1)}/\tau_{N-1}^{(1)}$ and $\rho'$ for $N > 2$.

Using $\tau_{21}^{\text{DDT}}$ as a single variable classifier on a 2-prong signal gives an AUC of 0.687, which is the lowest among the decorrelation methods considered in this work. Compared to $\tau_{21}$, the Designed Decorrelated Tagger has an AUC that is 8% lower, though only a nominally smaller background rejection at a fixed signal efficiency of 50%, as seen in the left panel of Fig. 5. The right panels of Fig. 5 show how the background distribution changes as tighter cuts are made on the signal efficiency. $\tau_{21}^{\text{DDT}}$ sculpts far less than $\tau_{21}$ (See App. C for a side-by-side comparison), and by eye, seems to perfectly preserve the shape of the QCD background distribution. We quantify these statements in the next sections.

### 3.2.2 Planing

Data planing [9] is a procedure that was initially designed to better understand what information an MV model is learning. This is accomplished by using the "uniform phase space" scheme introduced in [4] to restrict the model's access to a certain observable, and looking for a subsequent drop in performance during testing. It turns out, however, that limiting what information the neural network is capable of learning and decorrelating the network output

from a given observable are similar tasks.

At its core, planing is a weighting technique that takes a given distribution, and weights the data such that this distribution is now uniform over the range of values in consideration. Our choice to weight both the signal and the background to be uniform is not unique—one could instead weight the signal to the background shape or vice versa, as long as they have the same distribution after the procedure. For a set of input features, $X_i$, where $i$ denotes a given event, and $m$ is the feature to be planed, the weights can be computed as:

$$[w(X_i)]^{-1} = C \left. \frac{d\sigma(X_i)}{dm} \right|_{m=m_i}, \tag{6}$$

where $\sigma(X_i)$ is the distribution of the data as a function of feature $X$, and $C$ is a dimensionful constant common to both signal and background. This is required, as signal and background are planed separately. In practice, these weights are determined by uniformly binning the events, and then inverting the resulting histogram. This introduces some finite binning effects, which tend to be more pronounced near the ends of the distribution. However, these effects can be easily mitigated, and do not have a significant impact on training, see Ref. [49] for a method to compute the weights without binning.

The planed feature does not necessarily have to be an input to the network. In this work, we are interested in decorrelating the network output from the jet mass, so this is the variable we apply the planing procedure to. As mentioned in Sec. 3.1, it is possible to add event-by-event weights to the loss function when training, treating some events as more or less important than others. Planing uses the weights in Eq. (6) and treats events that weigh less (more) as more (less) important. When training a network on planed data, the weights in the binary cross-entropy, Eq. (5), are the product of the planing weights, Eq. (6), and the class weights discussed previously.

Figure 6 highlights the key features of planing. In the left panels, we show the jet mass distributions for the 2-pronged signal events and the QCD background events. These distributions are planed separately, and the lower left panel shows the resulting distributions after planing away the jet mass information. Both are uniform over the relevant mass range, though there are some finite binning effects visible near the high- and low-mass ends of the planed distributions. The two center panels show one of the network inputs, $\tau_1^{(1)}$, before and after planing. Before planing there is a clear separation between the signal and background distributions, which means that there is discriminating power available to the network from this feature alone. After weighting this input, we see that the signal and background $\tau_1^{(1)}$ look much more similar, so there is now less discriminating power in this planed feature. However, planing does not reduce the discriminating power of every input feature. In the rightmost panels, we see that before planing, the distributions of $\tau_2^{(1)}$ are nearly identical for signal and background. After applying the weights from the planing procedure, we see that there is now more distinction between the two, with the added benefit that this extra classifying power does not come at the cost of further sculpting the background jet mass distribution.

The MV classifier is trained on planed data, but is tested using unaltered data. Compared to a network with the same architecture, but trained on unaugmented data, the network trained on planed data is only able to achieve an AUC of 0.778—nearly 11% lower. This reduction in AUC corresponds to a background rejection nearly 3 times smaller at a fixed signal efficiency of 50% compared to the network trained on data which has not been planed, as seen in the left panel of Fig. 5. The right panels of Fig. 5 shows how the background distribution changes as tighter cuts are made on the signal efficiency. Comparing these distributions to the right panels of Fig. 4, it is clear that a network trained on planed data sculpts far less than any of the MV techniques discussed thus far. A side-by-side comparison can be found in App. C. We quantify these statements in the next sections.

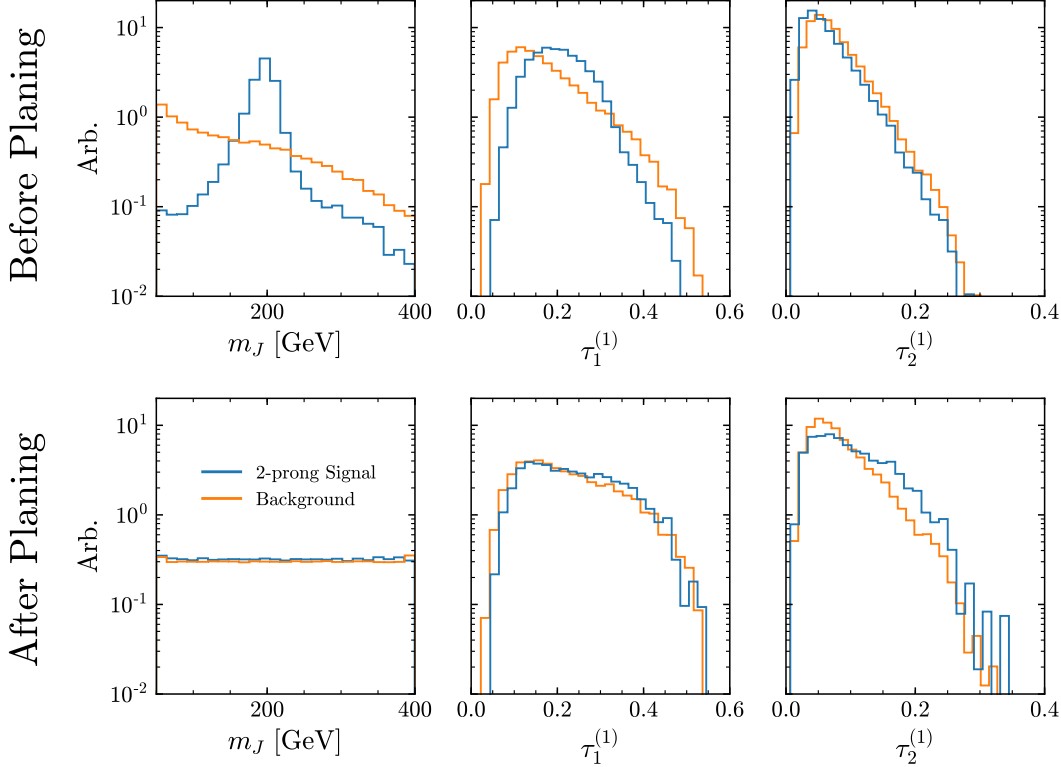

Figure 6: The upper and lower panels show distributions before and after planing away the jet mass, respectively. The left panels show the jet mass distribution for the 2-pronged signal and QCD background. By design, both distributions are (nearly) identical, and uniform across the entire mass range after planing. The center panels show $\tau_1^{(1)}$, one of the input variables for the classifiers. Before planing, this variable has discriminating power, but that was correlated with the jet mass and got removed by the planing process. The right panels show $\tau_2^{(1)}$, which has more separation between signal and background *after* planing.

### 3.2.3 PCA

Another preprocessing procedure which aims to decorrelate the discrimination power of the NN from the jet mass was proposed in [7]. The basic idea is to preprocess the $\tau_N^{(\beta)}$ variables in such a way that their distribution for QCD events is no longer correlated to the jet mass. This is achieved by first binning the standardized data (zero mean and unit standard deviation for each variable) in jet mass, with a variable binning size to have the same number of QCD events in each bin. Then, in each bin, the standardized input variables are transformed as follows:

$$\vec{\tau}_i^{\,\text{std}} \rightarrow \vec{\tau}_i^{\,\text{PCA}} = R_i^{-1} S_i R_i\, \vec{\tau}_i^{\,\text{std}}, \tag{7}$$

where $\vec{\tau}_i^{\,\text{std}}$ ($\vec{\tau}_i^{\,\text{PCA}}$) is a 11 dimensional vector made of standardized (PCA transformed) variables in bin $i$, $R_i$ is the matrix that diagonalizes the covariance matrix for the QCD $\tau$ variables in that given bin, and $S_i$ makes the covariance matrix unity in that bin. The action of $R_i$ is to induce a rotation into a basis where all the variables are linearly uncorrelated (this is the typical procedure used in principal component analysis (PCA), from which the method derives its name). Typically, after this rotation the data needs to be standardized again, requiring the action of the diagonal $S_i$ matrix. The effect of PCA preprocessing procedure is illustrated by the scatter plot in Fig. 7, for two of the variables $\tau_1^{(1)}$ and $\tau_2^{(1)}$, for three mass bins. In the

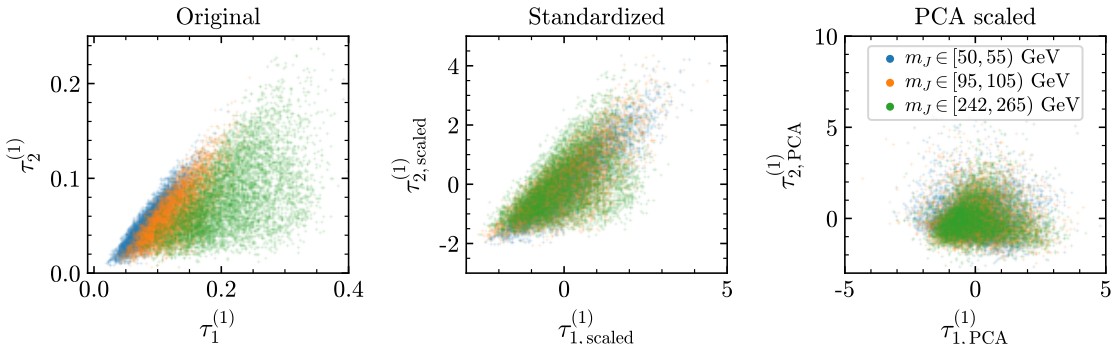

Figure 7: Scatterplot of two benchmark $\tau$ variables for QCD events in three differ-ent mass windows. The left panel shows the original variables, before any kind of preprocessing. The events from different mass bins are well separated. The center panel shows the same events after removing the mean and setting the variance of each variable in each bin to unity. The different mass bins now have the same range, but the 2D correlations are still distinct. In the right panel, the events have been standardized and PCA transformed on a linearly independent basis. The different mass ranges are now hard to distinguish.

scatter plot, the differences for the mass bins in the original variables are very easy to see, and also noticeable in the standardized variable. However, the mass bins look much more similar for PCA transformed variables. Notice that, while both the $R$ and $S$ are computed (bin-by-bin) only using the QCD sample, the transformation, Eq. (7), is then applied both to the QCD and signal events (both during the training of the NN and when applying the tagger to the test data).[6]

The network trained on PCA scaled data is able to achieve an AUC of 0.829, which is only a 4% reduction compared to the network with the same architecture trained on the unaltered data. This is shown in the left panel of Fig. 5. The right panel of Fig. 5 shows how the background distribution changes as tighter cuts are made on the signal efficiency. Comparing these distributions to the right panels of Fig. 4 (see App. C for the side-by-side comparison), it is again clear that a network trained on PCA scaled data sculpts less. We quantify these statements in the next sections.

## 3.3 Decorrelation based on training augmentation

The general idea of training augmentation is to assign a penalty to distorting a background distribution that is desired to be uncorrelated with the classifier. This allows the optimal solu-tion to balance the performance with decorrelation. Further, the decorrelation is not requested at just one step in the process, like in data augmentation based approach, but rather at each step in the process. In this category, we study two of recently proposed methods *uBoost* and *Adverserial Neural Networks*.

### 3.3.1 uBoost

A BDT algorithm can be modified to leave some distributions of a given class unaffected in the classification procedure, as proposed in [30], called uBoost.[7] The basic idea is to incorporate

---

[6]This is different than the case of planing, where the test set does not use data augmentation.

[7]A follow up to the uBoost algorithm was developed in Ref. [50]. This new method achieves similar classification and uniformity as uBoost, but only trains a single BDT with a modified loss function, rather than training mulitple BDTs.

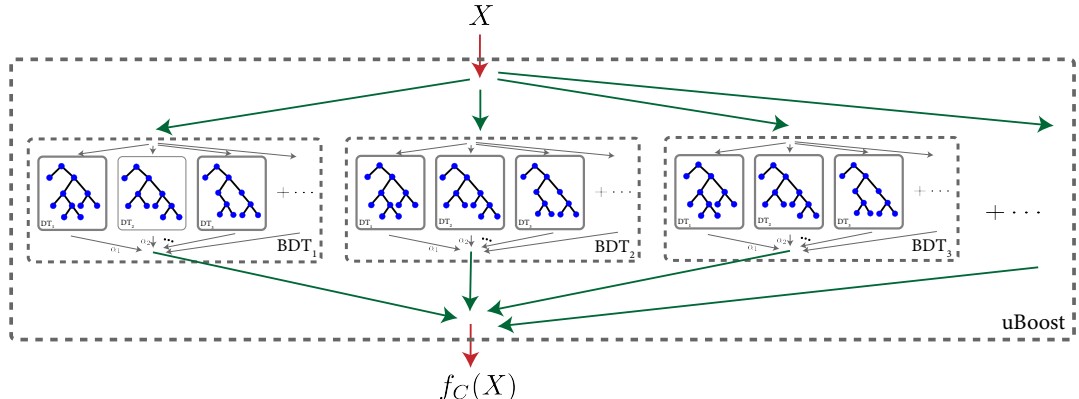

Figure 8: The network architecture used in the uBoost algorithm. Each BDT has the same layout as those in Fig. 2, and is tasked with keeping the background uniform at a given target signal efficiency. We use 20 BDTs to cover the entire efficiency range, and results are interpolated between target efficiencies to keep the background uniform over the whole efficiency range. The Gini index is used to measure the quality of a split, and the best split is taken when creating new branches.

the cost of affecting the distribution that is desired to be unaffected in the optimization procedure. This procedure necessarily depends on the efficiency of classification, since the cost of affecting a distribution has to be measured for fixed efficiency. In other words, a trivial way to not affect a distribution for a variable for a given class is to have a very small efficiency to select the other class, so that no events of the other class are selected and the distribution stays the same. Hence, the non-trivial optimization algorithm is implicitly defined for a given efficiency, taken to be the average efficiency of the BDT. The average efficiency of the overall BDT corresponds to a local efficiency for each event. This local efficiency is calculated using k-nearest-neighbor (kNN) events that pass the BDT cut, constructed from DTs up to this point. Hence this local efficiency depends on both the event and the tree. Data points with a local efficiency lower than average efficiency are given more importance, and those with a local efficiency higher than average efficiency are given lesser importance. The relative importance is controlled by a parameter $\beta_u$ (see Eq.(2.3) in Ref. [30]). The BDT then is optimized for a given efficiency. One can then construct an even bigger ensemble of BDTs, each optimized for a given efficiency, and design the response function in such a way that the right one is chosen for a given efficiency. An illustration of this is sketched in Fig. 8.

The uBoost architecture we consider uses 20 BDTs to cover the full signal efficiency range, with each BDT being comprised of 150 individual DTs, each with a maximum depth of 4. The decision trees use the Gini Index to measure the quality of a split. Additionally, we use $k = 50$ nearest neighbor events to compute the local efficiencies. As the authors of [30] point out, there is very little change in the performance of uBoost for $k \in [50, 1000]$, but choosing $k < 20$ drastically increases the statistical uncertainty on the local efficiency, which worsens the performance of the uBoost algorithm. The parameter $\beta_u$ which sets the relative training importance of events with local efficiency more/less than the average efficiency, is set to 1.

Using the uBoost algorithm for classification results in an AUC of 0.783, which is a 9% reduction when compared to classification using standard gradient boosted decision trees. At a fixed signal efficiency of 50%, this translates into uBoost rejecting 23% less background than a standard BDT operating at the same signal efficiency. However, this reduction in classification power comes with the benefit of decreased background sculpting. The right panel in Fig. 9 shows how the background distribution changes as tighter cuts are made on the uBoost net-

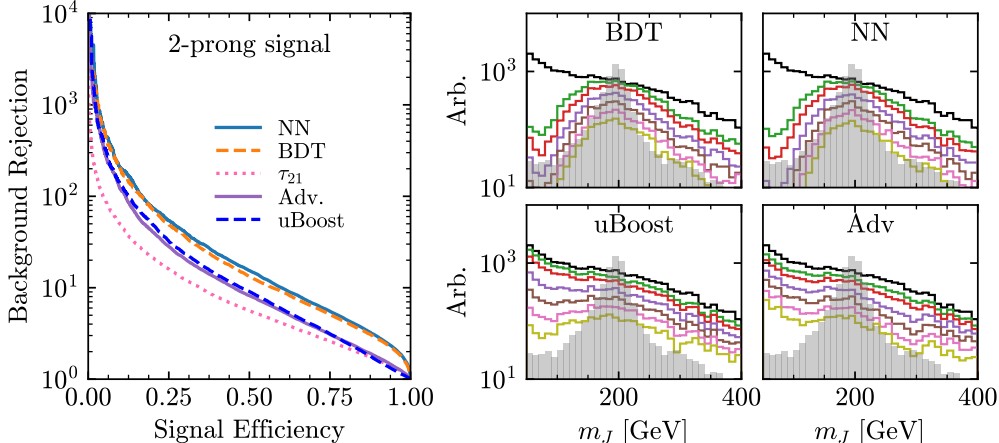

Figure 9: The left panel shows the ROC curves for the adversarially trained neural network and uBoost, along with the results of the base neural network and $\tau_{21}$, for comparison. The adversarial results use $\lambda = 50$, and the uBoost results use $\beta_u = 1$. The right panels show the *background only* distributions as successively tighter cuts are made on the output of these classifiers: signal efficiency of 1.0 (black), 0.95 (green), 0.9 (red), 0.8 (purple), 0.7 (brown), 0.6 (pink) and 0.5 (yellow). The full 2-pronged signal is shown in the filled-in grey distribution for context. Both these methods are able to preserve the background shape well, with only a marginal decrease in performance, but take a factor of 10 to 100 more time to train. Compared to their MV counterparts in the upper panels, it is clear that the training augmentation based approaches significantly reduce the extent of the background sculpting. A full side-by-side comparison for 2, 3, and 4 prong signals is shown in App. C.

work output. By eye, uBoost sculpts the background considerably less than a traditional BDT. Quantitative assessments are made in Section 4.

### 3.3.2 Adversarial

The idea to use adversarial networks to decorrelate jet mass from the output of a classifier was first introduced in [27]. The authors showed that in the case of small systematic errors, both adversarially trained networks and traditional neural networks lead to better chances of discovery for 2-pronged jets than using traditional jet substructure or the DDT [28]. However, when the systematic uncertainty on the background is large, the traditional neural network never does as well as the adversarially trained network or the analytic taggers. The adversarially trained network remains better than the analytic methods.

The key aspect of adversarial training is using multiple neural networks, instead of single one. First, the inputs are fed through a traditional classifier, as in Sec. 3.1. The output of the classifier is a number between 0 and 1. The next stage trains a second network to infer the feature to be decorrelated (the mass for us) using only the output of the classifier. An illustration of this is shown in Fig. 10.

The overall goal then becomes to train a classifier which not only classifies well, but which also does not allow the adversary to infer the jet mass. This is done using a combined loss function of the form

$$L_{\text{tagger}} = L_{\text{classifier}} - \lambda \, L_{\text{adversary}}, \tag{8}$$

where $L_{\text{classifier}}$ and $L_{\text{adversary}}$ are usual classification loss functions. However, we only calculate $L_{\text{adversary}}$ for the QCD sample and not the signal samples. The parameter $\lambda$ is a positive

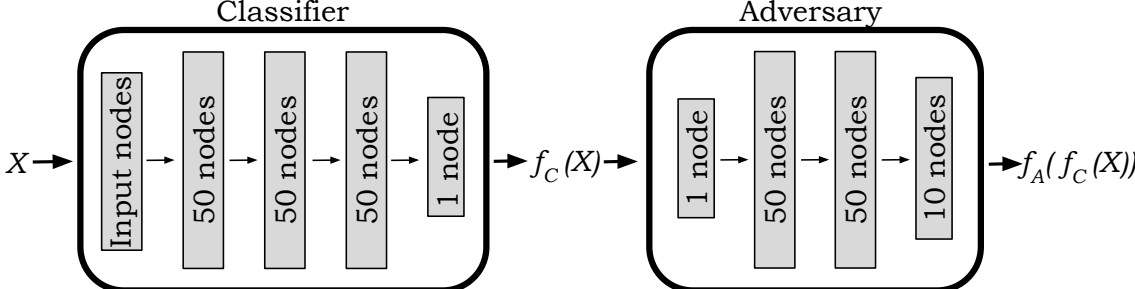

Figure 10: The setup of our adversarially trained neural network. The classifier has the same hyperparameters as in Fig. 3. The output of the classifier becomes the input of the adversary, which attempts to predict which bin of the jet mass the QCD events came from. We use tanh activation for the hidden layers of the adversary, and softmax activation for the final layer, with 10 outputs. The multi-class cross entropy loss function is used for the adversary.

hyperparameter set by the user, giving the relative importance of the two tasks; classifying and decorrelating. A larger value of $\lambda$ puts more emphasis on not allowing the adversary to be able to infer the mass at the cost of poorer classification.

As done in Ref. [27], we use ten nodes for the output of the adversary, with the jet mass digitized to ten bins with equal numbers of QCD jets per bin, treating the problem as a multi-class classification problem. The activation for the last layer is the softmax function and $L_{\text{adversary}}$ is the multiclass cross entropy. This was found to lead to more stable training than trying to regress the exact jet mass. In addition, we found that a tanh activation function for the hidden layers of the adversary to be more stable than ReLu activation. The ATLAS study in Ref. [33] also uses adversarial neural networks for mass decorrelation, but does so by having the adversary predict the probability distribution function of the background, as in Ref. [31], rather than predicting the mass bin.

The adversarial set-up makes training the networks more involved. First, we train the classifier using only the binary cross entropy loss function. Next, the adversary is trained alone, only using the output of the classifier. We found the training procedure which led to the most stable results for the combined networks to be as follows. The adversary is set to not be trainable, and the classifier weights are updated using the total loss of Eq. (8). However, only a small number of updates to the weights of the classifier are allowed. Then, the classifier weights are frozen and the adversary becomes trainable. It is given substantially more time to adjust to the updated classifier, minimizing its own $L_{\text{adversary}}$ for many epochs. The process is then repeated many times, first making minor updates to the classifier followed by ample time for the adversary to respond. This procedure takes about a factor of 10-100 more time to train than other methods.

The other aspect of adversarial training which makes it more challenging is the choice of the hyperparameter $\lambda$. *A priori*, the value of $\lambda$ should be chosen so that the loss of the classifier is of order the same size as the loss of the adversary. However, the best value will depend on the use case. The necessity of this optimization produces a family of classifiers with trade-offs between classifying power and decorrelation abilities. This is in contrast to analytic and data augmentation based decorrelation methods, which only give a single classifier. For our studies, we scanned over ranges of $\lambda \in \{1, 2, 5, 10, 20, 50, 100, 200, 500, 1000\}$. The results seem to saturate at $\lambda = 50$. The result of this hyperparameter scan are shown in App. A. We tried smaller values as well, but these were seen to be nearly equivalent with the traditional neural network. The longer training times, coupled with the need to optimize $\lambda$ greatly increases the computational overhead for using adversarial methods.

The adversarially-trained neural network (with $\lambda = 50$) achieves an AUC of 0.807, which is a 7% reduction in AUC compared to the neural network considered in section 3.1. At a fixed signal efficiency of 50%, this difference in AUC translates to the adversarially trained network rejecting 33% less background than a traditionally trained neural network. However, the adversarial approach still results in a better classifier than single variable analyses, as shown in Fig. 9. The right panel of Fig. 9 shows how the background distribution changes as tighter cuts are made on the output of the adversarially trained network. It is clear that the adversarial approach sculpts the background far less than traditional neural networks. We make this statement more quantitative in Sec. 4.

## 4 Results

One of the considerations when choosing an analysis method is the computational overhead. Table 2 shows the amount of time it takes to train the different classifiers. The difference between the number of prongs is mostly dominated by the different sample sizes, but also comes from how easy the minimum of the loss function is to find.

The neural network based methods take longer to train than the boosted decision trees. As expected, the methods which augment the training process take longer to return a good classifier. The uBoost method trains 20 different BDTs so it takes around 20 times longer than the base BDT.[8] Decorrelating the NN by using an adversary network takes substantially longer to train, although as we show below, it does achieve the best results. In contrast, the methods which augment the data beforehand show very little change in the time it takes to train.

The computational overhead is not the only consideration. In the rest of this section, we examine both the amount of background rejection and the degree to which the background is sculpted. Depending on the particular analysis, it may be optimal to allow more or less sculpting depending on the needed background rejection. The background rejections is defined over the whole jet mass range considered: $50 \leq m_J(\text{GeV}) \leq 400$. For the taggers considered in this work, we expect this choice to give qualitatively the same results that would be obtained by defining it in a narrower mass window centered around the signal. This is because they are structured exactly to achieve this goal: to keep the background rejection constant over the whole mass range.

To quantitatively define how much the classifier sculpts the background, we use the Bhattacharyya distance, which is a popular measure of the distance between two probability distributions. For two given histograms $H_1$ and $H_2$ with $N$ bins each, the distance is given as:

$$d_B(H_1, H_2) = \sqrt{1 - \frac{1}{N\sqrt{\langle H_1 \rangle \langle H_2 \rangle}} \sum_I \left( \sqrt{H_1(I)H_2(I)} \right)}, \qquad \langle H_K \rangle = \frac{1}{N}\sum_J H_K(J). \qquad (9)$$

This distance has the nice property that it is normalized between 0 and 1, allowing for a comparison of the sculpting from various taggers more easily. This choice of metric is not unique. In App. B, we compare the Bhattacharyya distance with another distance measure, the Jensen-Shannon distance (used in Ref. [33]). The two are seen to have similar features.

### 4.1 Augmented training

In this section we examine the decorrelation methods which change the way the training is done, namely the adversarial neural networks and uBoost. While these methods take longer

---

[8]The updated boosting methods found in [50] do not require training multiple BDTs, so their training time is similar to a standard BDT.

Table 2: The time in seconds to train a classifier on dual E5-2690v4 (28 core) processors. The mean and standard deviation are calculated over 10 independent trainings. The large variance in the neural network times is due to the early stopping condition, leading to a non-fixed number of epochs. Note that the adversarially trained neural network statistics are over sampled once over each of the nine different values of $\lambda$ due to the long training time. In addition, the adversarial networks used GPU nodes. BDTs are faster to train, but are not as effective classifiers. The Adversarial and uBoost decorrelation methods take much longer than the PCA or Planing methods.

| Method | 2-prong | 3-prong | 4-prong |
|---|---|---|---|
| Base Network | $409 \pm 56.8$ | $601 \pm 82.9$ | $483 \pm 64.9$ |
| Base BDT | $66 \pm 2.7$ | $88 \pm 0.4$ | $64 \pm 1.1$ |
| PCA Network | $421 \pm 48.7$ | $566 \pm 63.6$ | $366 \pm 32.8$ |
| PCA BDT | $70 \pm 1.3$ | $97 \pm 1.3$ | $69 \pm 0.9$ |
| Planed Network | $406 \pm 44.2$ | $604 \pm 90.7$ | $462 \pm 81.7$ |
| Planed BDT | $64 \pm 1.0$ | $88 \pm 1.2$ | $63 \pm 0.8$ |
| Adversarial | $49\,429 \pm 520.8$ | $54\,953 \pm 683.3$ | $49\,003 \pm 1892.0$ |
| uBoost | $1495 \pm 6.6$ | $2047 \pm 6.5$ | $1430 \pm 10.0$ |

to train, their input data is unaltered, which is better for calibration and other systematics. In all of the comparisons, we include the base neural network and the single-variable analysis as benchmark references.

Figure 11 shows the ROC curves for decorrelation methods along with the benchmarks. The left, middle, and right columns are for the 2-prong (boosted $Z_{KK} \rightarrow q\bar{q}$), 3-prong (boosted top), and 4-prong (boosted $R \rightarrow q\bar{q}q'\bar{q}'$) jets as described in Sec. 2, respectively. The first noticeable trend is that the more prongs the signal sample contains, the easier it is to distinguish from the QCD background, which is typically single pronged. In fact, for many of our classifiers for the 4-prong signal, we run out of background events at a signal efficiency of around 0.1. We will see evidence of this in the remaining metrics even though the rapid removal of background events yields more statistical uncertainty on these results.

The adversarially trained network with $\lambda = 50$ is shown in the solid light-purple line and uBoost classifier is shown by the dashed blue line. This value of $\lambda$ was around where the performance saturated; Appendix A shows the results for all values of $\lambda$ tested. For the 2-prong signal, uBoost and the adversarially trained network have very similar curves. These are roughly in the middle of the base MV methods and the single variable analysis. Moving to the 3- and 4- prong signals, the adversarially trained network achieves better background rejection than uBoost, but both of these are significantly better than a single variable analysis. Note that currently there are no DDT type methods for 3- and 4-prong jets.

The Bhattacharyya distance calculated on the *QCD background only* distributions is shown in Fig. 12. Specifically, we calculate the distance between the original (no cuts) jet mass distribution and the background distribution which passes a cut for the specified signal efficiency (top row) or background rejection (bottom row). We see clearly that the original NN and BDT

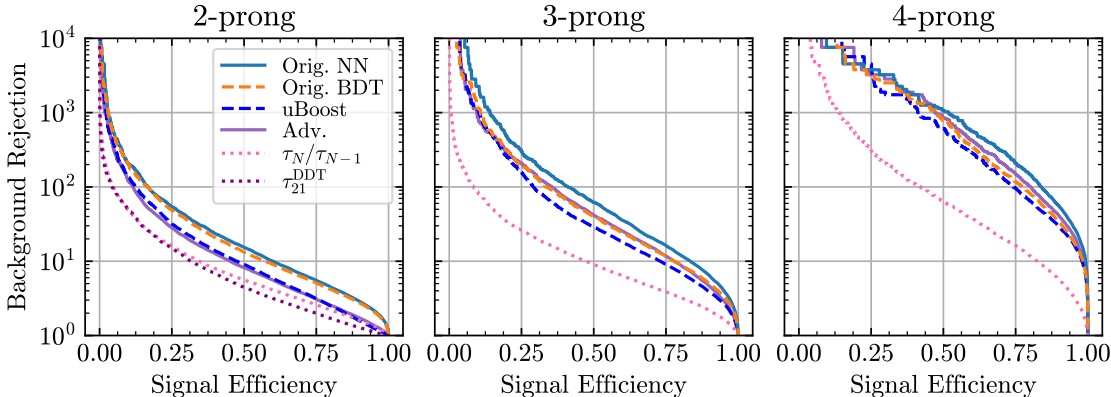

Figure 11: ROC curves for the 2-, 3-, and 4-prong signal jets versus QCD background for the methods which augment the training method to decorrelate the jet mass. The solid, dashed, and dotted curves show results for neural networks, boosted decision trees, and single variable analysis, respectively. The light blue curves are for the traditional method benchmarks. The purple and dark-blue lines denote the adversarially trained network and uBoost decision tree. For the 3- and 4-prong cases, uBoost cannot classify as well as the adversarially trained neural networks, but still does much better than using a single variable, $\tau_3/\tau_2$ and $\tau_4/\tau_3$, respectively

give the greatest amount of distortion to the distributions, resulting in larger distances. For the 2-prong jets, the distance for original MVs is around 0.5 for most of the signal efficiencies, and $\tau_{21}$ slowly grows to the same values. For 3- and 4-prong, the single N-subjettiness variable produce smaller distances than the original MVs over the whole region.

The $\tau_{21}^{\mathrm{DDT}}$ classifier was specifically designed to remove the mass correlation; as such, it produces the smallest distances for fixed signal efficiency. However, there are no 3- or 4-prong versions. That being said, the adversarially trained neural network produces distances that are comparable to $\tau_{21}^{\mathrm{DDT}}$ over the range of signal efficiencies. It also has the smallest distances for the MV methods for the 3- and 4-prong signals. uBoost does not achieve as low of distance scores but its distances are still generally closer to the adversarially trained network than the originals, and trains about a factor of 30 faster than the adversary.

Only looking at the distance compared to the signal efficiency does not take into account how well the classifier separates the signal jets from QCD. Balancing the need for unaltered distributions against the necessary background rejection is task specific, but can be aided by plotting the two against each other. In the lower row of Fig. 12, we show the parametric plots of the histogram distance versus the background rejection. In these plots, the optimal classifier will be to the lower-left corner, yielding a small distance between the distributions before and after cuts and simultaneously rejecting large backgrounds. These are made by scanning over the values of the signal efficiency from 1 to 0.05, which is why the curves do not extend all the way to the left. The points marked by circles, stars, and squares are for fixed signal efficiencies of 0.75, 0.5, and 0.25, respectively.

The original MV methods, along with the single variable analysis, yield similar shaped curves, offering the same amount of sculpting for a fixed amount of background rejection. This is interesting because the $\tau_N/\tau_{N-1}$ distances were quite different when plotted against the signal efficiency. This can be observed by examining the location of the marked points along the curve, where the pink ones fall further to the left than do the light blue and orange points.

The adversarially trained classifier sculpts the least for a given background rejection for

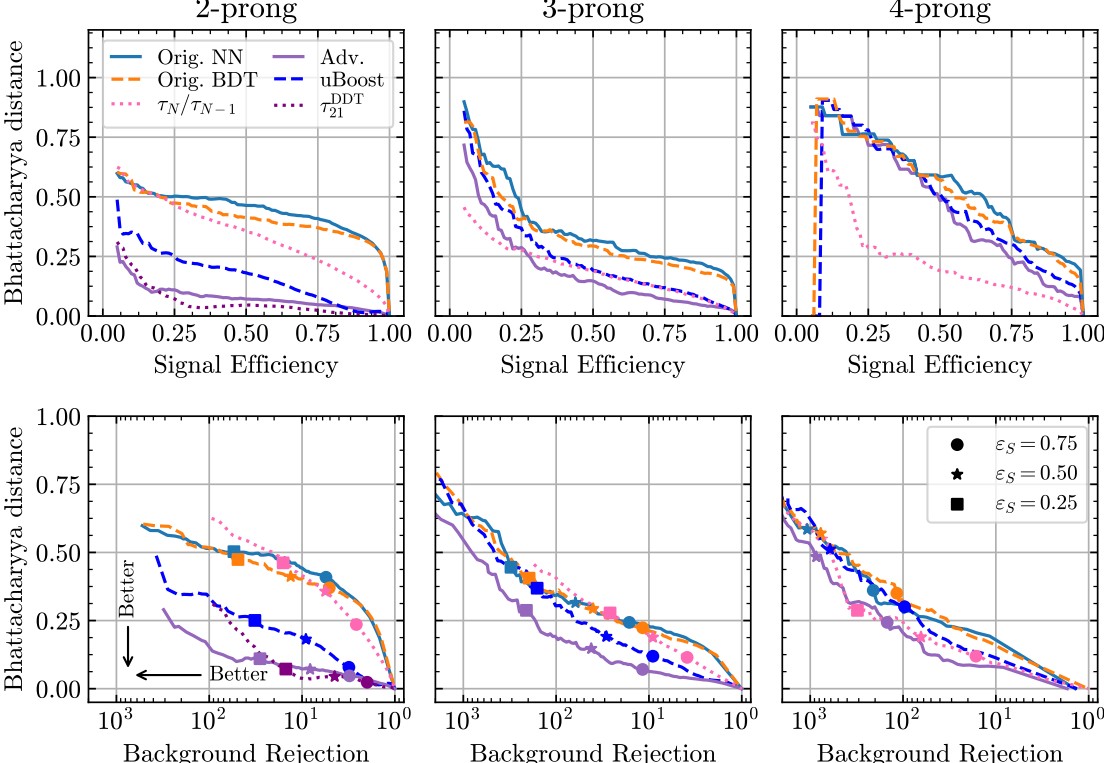

Figure 12: The Bhattacharyya distance for the QCD background distributions compared to the original distributions. The distance is defined in Eq. (9), and a larger distance represents more sculpting—lower on the plot is better. The upper and lower rows plot the distance as a function of signal efficiency or background rejection, respectively. $\tau_{21}^{\mathrm{DDT}}$ produces the smallest distances for fixed signal efficiency, but does not generalize to higher-prong jets. The adversarially trained network yields a close approximation and generalizes to more prongs. uBoost falls between the original methods and the adversarially trained network, but takes a factor of 30 less time to train.

the different pronged jets, other than a small region where $\tau_{21}^{\mathrm{DDT}}$ is the least. uBoost again falls between the original methods and the adversarially trained network, providing a good compromise on computation time and decorrelation.

For the 4-prong jets, all of the classifiers give similar results with fairly large distances. This indicates that the QCD is not 4-pronged, so all of the classifiers can cut out large amounts of the background. Even the methods which are supposed to produce smaller histogram distances end up sculpting the backgrounds quite heavily. In any real analysis, this is most likely not an issue because of the extensive background rejection.

Plotting the distance versus signal efficiency (top row of Fig. 12) makes it hard to see trends in sculpting between the various pronged jets. However, in the bottom row, we get a sense that the decorrelation techniques yield a certain distortion of the background shape given the amount of rejection. For instance, with a background rejection of 10, $\tau_{21}^{\mathrm{DDT}}$, uBoost, and adversarially trained networks yield Bhattacharyya distances $\sim 0.1$ for all of the prongs. Additionally, the distance is $\sim 0.25$ for a background rejection of 100 for all prongs. This is expected because our different pronged signal distributions peak at roughly the same mass (200 GeV for 2- and 4-pronged, and 173 GeV for 3-pronged). Thus, for a fixed background rejection, the background events which remain mimic a signal region that is approximately

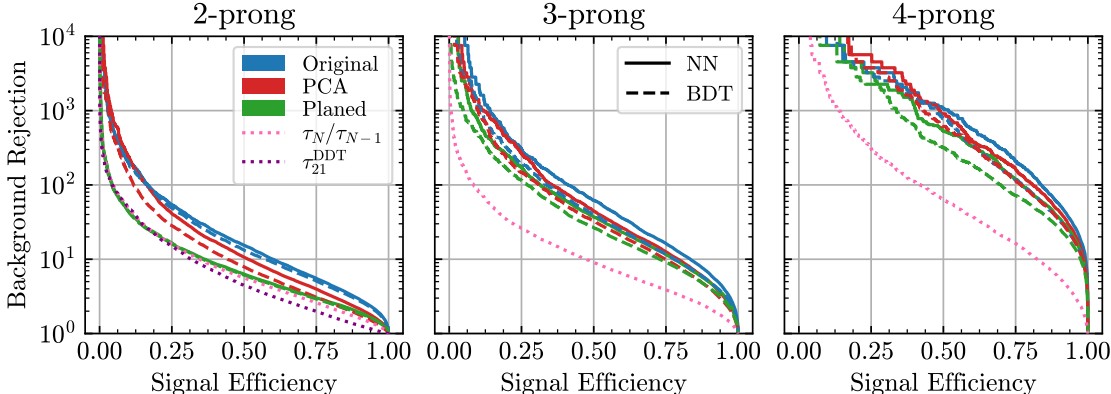

Figure 13: ROC curves for the 2-, 3-, and 4-prong signal jets versus QCD background for the methods which augment the data to decorrelate the jet mass rather than augment the training. The dashed and solid lines show the gradient boosted decision trees (BDT) and neural networks (NN), respectively. The blue, red, and green curves are for the data which has not been altered, data which uses the PCA rescaling, and data which has the jet mass planed away. The dotted lines show the results using a single combination of the N-subjettiness variables. Generally the BDTs have slightly worse background rejection than the NNs. Similarly, the PCA rescaling based methods tend to be between the unaltered methods and the planing methods, which are better than the single variable analyses.

independent of the signal prongedness.

## 4.2 Augmented data

The previous section examined the extent to which uBoost and adversarially trained neural networks can decorrelate the jet mass from the classifier output, which is achieved by changing the training procedure. We now move on to focus on the methods proposed in Sec. 3.2: altering the input data rather than the training. Augmenting the data rather than the training procedure greatly reduces the amount of time required to train the models, as shown in Tab. 2. Additionally, it allows us to test the methods using both boosted decision trees and neural networks.

The overall ability to classify is shown in the ROC curves in Fig. 13. As with the last section, the left, middle, and right plots have the signal jets with boosted two-body, three-body, and four-body decays, respectively. In all of the plots, the blue, red, and green lines are for the unaltered data, the PCA rotated data, and the Planed data respectively. The solid lines represent the neural network results, and the dashed lines are the gradient boosted decision tree. Additionally, we show the single N-subjettiness variable analyses in the dotted lines.

In all of the plots, the unaltered neural network achieves the best classification. This is expected, because neural networks can use more non-linearities, and the data has not been processed to remove correlations with the jet mass. The 2-prong signal shows some difference in the PCA and Planed neural network results, but for the 3- and 4-prong signal neural nets, these methods yield similar classification. The BDTs show similar trends, performing slightly worse than the neural networks in terms of pure classification. The methods to decorrelate the jet mass from the MV output still achieve better background rejection than the single variable analysis.

The degree of decorrelation is examined in Fig. 14 where the Bhattacharyya distance is plotted against the signal efficiency in the upper row. The distance is calculated on the

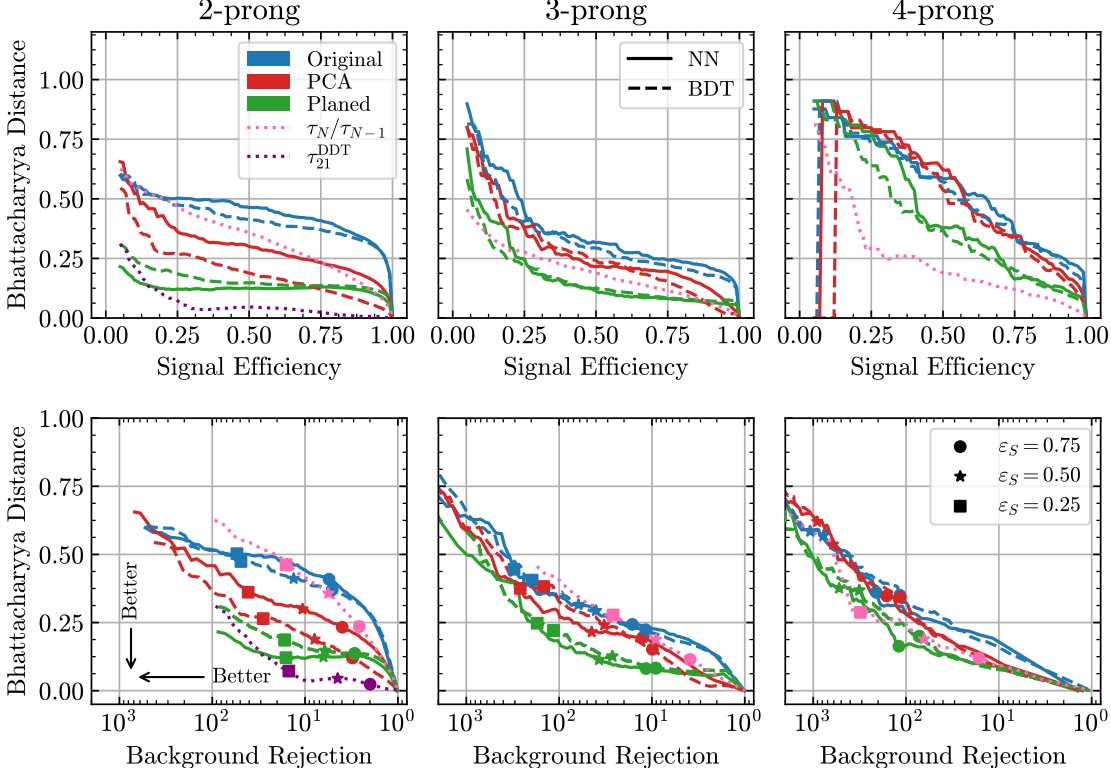

Figure 14: The Bhattacharyya distance for the QCD background distributions compared to the original distributions. The distance is defined in Eq. (9), and a larger distance represents more sculpting—lower on the plot is better. The neural networks tend to sculpt the distributions worse than the BDT, regardless of the data. Both the PCA rotations and Planing the jet mass result in smaller distances than the classifiers trained on the original data.

background-only distributions and the color scheme is the same as the previous figure. In almost every case, the BDT has smaller distances (less distortion) than the NN. The classifiers trained on the PCA rotated data show much less distortion than the original data other than for the 4-prong jets. For instance, the 2-prong jet mass distribution distances are about half the value as the corresponding unaltered method. The method of planing away the jet mass information shows nearly an additional factor of two less sculpting than the PCA method for the 2-prong jets. However, the planing curves do not reach as low of distances as $\tau_{21}^{\mathrm{DDT}}$ for most signal efficiencies.

The planing method produces the smallest distances out of the different methods considered here for the 3-prong jets. The 4-prong signal is particularly easy for the classifiers to distinguish from the QCD background. As a result, even the MVs with attempts at mass decorrelation have large Bhattacharyya distances for fixed signal efficiency. Out of these, the planing method sculpts the distributions the least.

In the bottom row of Fig. 14 we again show the background rejection plotted against the Bhattacharyya distance. We again find that for 2-prong jets, $\tau_{21}^{\mathrm{DDT}}$ sculpts the least for a given background rejection. However, it does not reach the largest background rejection values. The next best method is the neural network trained on planed data, which even produces smaller distances for background rejection above around 20, as compared to $\tau_{21}^{\mathrm{DDT}}$. The planing methods seem different than the others in that the NN has less sculpting than the BDT. The BDT trained on the PCA scaled data behaves similar to the BDT trained on planed data, but

reaches to larger background rejections and for a fixed background rejection has better signal efficiency. The PCA scaled neural network has slightly more sculpting for fixed background rejection than the other decorrelation methods, but still has much smaller distances than the unaltered methods.

The 3-prong jet signal Bhattacharyya distance shows an interesting change when plotted against the background rejection as opposed to the signal efficiency. In the middle panel of Fig. 14, $\tau_3/\tau_2$ produces smaller distances for fixed signal efficiency than all of the methods other than planing. However, for a fixed background rejection, it sculpts the data more than nearly all of the MV methods. We again find that the neural network trained on planed data provides the smallest distances for a given background rejection, but the BDT is not far behind. The PCA-based methods also provide less sculpting than the original methods.

The 4-prong jet results are more clustered, but $\tau_4/\tau_3$ (shown in pink) has smaller distances for fixed background rejection than the original methods—and surprisingly—the PCA based methods. That being said, the signal efficiencies are also much smaller. The neural network trained on data which has had the jet mass planed away produces the best curve.

The data augmentation methods explored in this section allow for using both BDTs and NNs and training takes about the same amount of time as the unaltered data. However, by augmenting the data, it is possible to make the MVs sculpt the jet mass much less than the original MVs. This does lower the overall background rejection for a given signal efficiency, but for fixed background rejection, the degree of sculpting can be much less. In this regard, these methods achieve similar results to the methods which augment the training process instead of the input data which have already been studied in the literature.

### 4.3 Comparison

Finally, we want to get a sense for how the augmented training methods perform, as compared to the data augmentation methods. In Fig. 15 we show the Bhattacharyya distance versus the signal efficiency (top) and background rejection (bottom) for *only* the decorrelation methods and not the original methods. We only show the neural networks for the data augmentation methods because they achieve better background rejection than BDTs, for fixed signal efficiency. For 2-prong jets, $\tau_{21}^{\mathrm{DDT}}$ has the least sculpting for background rejections smaller than around a factor of 10, but for larger than this, the adversarially trained network has the smallest distances. The network trained on planed data has the next smallest distances for large background rejection. While the green line is close to the purple adversary line, the marked points are further to the right, indicating that the planed network does not have as much signal efficiency for the corresponding background rejection/histogram distance. However, it is worth pointing out that planing sculpts less than uBoost, and takes about a factor of three less time to train. For the 2-prong jets, the PCA based method sculpts the most out of the decorrelation methods. PCA, however, seems to perform far better when paired with BDTs rather than NNs. Comparing Figs. 14 and 15, we see that augmenting the data using the PCA approach and then training a BDT—as opposed to a NN—sculpts just about the same as uBoost does for fixed background rejections, but takes less than 1/20 of the time to train.

The 3- and 4-prong jets show similar patterns in their results. As emphasized before, there is currently not an analytic decorrelation method similar to $\tau_{21}^{\mathrm{DDT}}$ for higher prong jets. The neural networks trained on the data with the jet mass planed away achieve very similar curves to the adversarial network curves—and train about a factor of 100 times faster. One may worry that this is a sign that the adversary is not actually doing well for the higher pronged jets. In App. C we show the jet mass distributions and do not think this is the case.

With these higher-pronged jets, the PCA based rotation method gives similar curves to uBoost. However, the PCA method has two benefits over uBoost. First, the marked points are further to the left, indicating that for fixed signal efficiency, the PCA networks have more

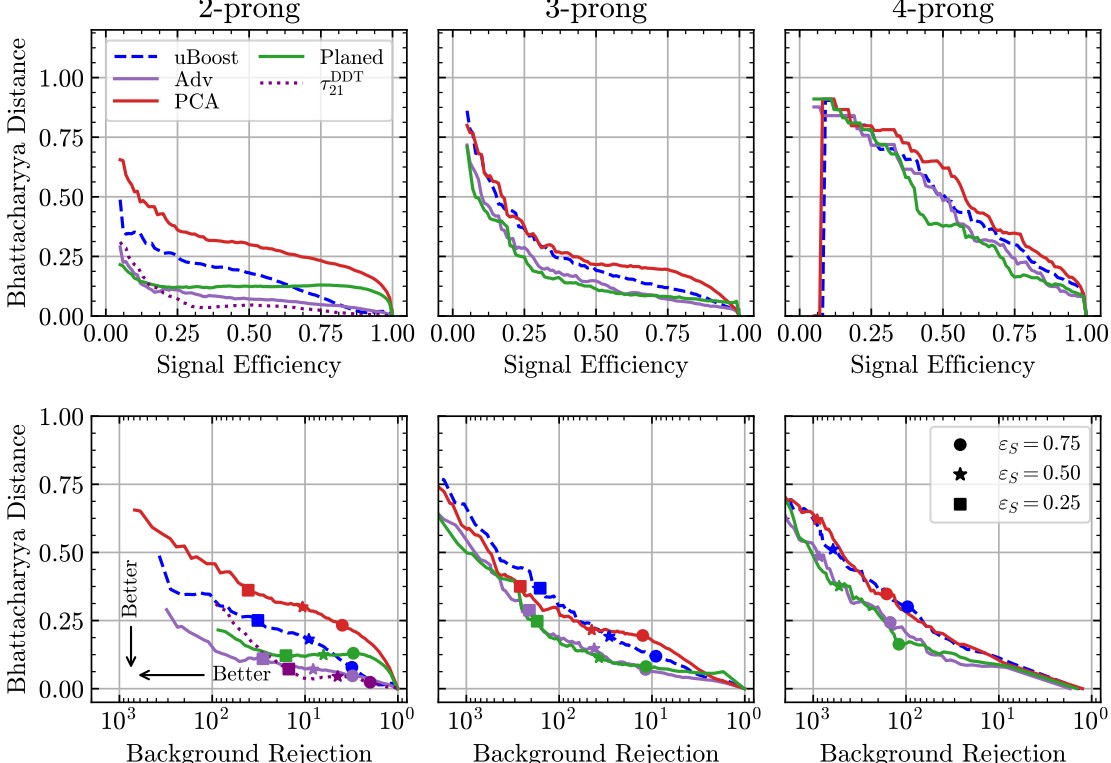

Figure 15: A comparison of all the MV based methods to decorrelate the jet mass from the classifier output. The shown PCA and Planed results are for NN architecture. The analytical $\tau_{21}^{\text{DDT}}$ method sculpts the least for moderate background rejection, but for larger values does not do as good as the adversarially trained neural network. The network trained on data augmented by planing the jet mass do almost as good as the adversarially trained network, with uBoost and the PCA based networks showing slightly more sculpting. With more prongs, planing and adversaries are nearly identical to each other while PCA and uBoost are very similar to each other.

background rejection than uBoost. Second, the amount of time required to train the machine is around a factor of four less.

## 5 Outlook and Future Work

The significance of a discovery or exclusion will always be the primary factor when determining which decorrelation method to use in an analysis. With this in mind, one potential concern with the data augmentation techniques is that they may reduce the statistical power of the data itself, especially when applied to data on the tails of the distribution. We do not expect this to be the case since the PCA based approach involves only linear transformations of the data and Planing only requires that the data used for training be reweighted. In future work, we will answer this question concretely, as part of a larger study where we explicitly look at a phenomenological proxy for the discovery significance that properly accounts for systematic errors and further optimizes the working point of the tagger.

One interesting extension of the techniques explored here would be to train networks on jets in a given mass range, and then use these networks to classify jets in an entirely different mass range. Neural networks offer a very flexible framework to train a wide variety of models,

but are far less adaptable once trained. The techniques studied here distinguish signal from background with less reliance on the jet mass. Since they only rely on substructure information, and not the absolute scale of the jet, they should be applicable to other regions of the mass parameter space. Showing that such results are possible could increase the usage of such MV techniques in large experimental collaborations, such as those at the LHC.

Our comparisons used the same representation of the input data for all of the classifiers, namely the N-subjettiness basis. However, there have been many studies of jet taggers using other representations, such as images, sequences, or graphs. Mass decorrelation has been done in images with Planing [4] and Adversarial training [32], but it would be interesting to see how all of the techniques studied here could be applied to the different representations, and if any additional advantage is offered. Additionally, decorrelating in both the jet mass and the transverse momentum could make for a stable jet tagger (See Ref. [49] for multidimensional decorrelation with Planing).

In this work, we applied all our methods to decorrelate the classifiers from the jet mass by explicitly using the jet mass in the decorrelation procedure (flattening the jet mass distribution for Planing; binning in jet mass for PCA). However, $\tau_{21}^{\text{DDT}}$ uses $\rho = \log(m^2/p_T^2)$ in its analytic decorrelation. An interesting test would be to examine how the decorrelation techniques work using this value (or just $p_T$) as opposed to the $m_J$ alone. Additionally, it would be worthwhile studying how robust these techniques are in a more realistic experimental environment by testing how the classification and decorrelation generalize to signals with mixed prongedness, and signal contamination. This is work we intend to do, and leave to future study.

Code to reproduce our results can be found on GitHub.

# 6    Conclusion

New physics searches are challenging, especially when the processes are rare and the backgrounds plentiful. Rejecting background events is necessary, but *how* the background is removed is also important. Experimental efforts to look for new physics are greatly aided by easy-to-model backgrounds, so the need for techniques that preserve the profiles of the underlying background distributions cannot be understated.

In this work, we explored a variety of cutting-edge methods used in the classification of boosted objects. We started by looking at how standard single- and multi-variate techniques achieve better classification at the cost of increased background sculpting. These standard methods serve as a point of comparison to analytic [28] and multivariate [27,30] methods designed specifically with mass decorrelation in mind. Previous studies of these techniques [33] focused only on their application to searches for two-body hadronic resonances. We extended these analyses to see how existing methods perform when tasked with classifying jets with more complex substructure. We also studied two data augmentation based techniques to decorrelate the classifier output from the mass of the jet, Planing and PCA-based rescaling, as well as two training augmentation based techniques, uBoost and Adversarial NNs.

All of the decorrelation techniques studied in this work reduce the extent to which the background is sculpted, and could therefore be used to increase sensitivity in a new physics search. We have shown that Planing and PCA give comparable performance to training augmentation based methods, while taking only a fraction of the time and computational overhead to train. These data augmentation techniques could be useful in situations such as testing prototypes, where fast turnaround is desired.

# Acknowledgements

We thank the 2018 Santa Fe Summer Workshop in Particle Physics where this project was started. BO also thanks the Munich Institute for Astro- and Particle Physics (MIAPP) for their hospitality, where much of this manuscript was written. RKM thanks Kavli Institute for Theoretical Physics (KITP) for hospitality where part of this work was finished. The authors thank Spencer Chang, Timothy Cohen, Jack Collins, Raffaele D'Agnolo, Gregor Kasieczka, Cris Mantilla, Ben Nachman, David Shih, and Nhan Tran for useful discussions and comments on the draft. BO is supported by the DOE under contract DE-SC0011640. This research was supported in part by the National Science Foundation under Grant No. NSF PHY-1748958.

# A  Adversary decorrelation parameter

As mentioned in Sec. 3.3.2, adversarially-trained neural networks introduce a new positive hyperparameter, $\lambda$, which must be chosen by the user. Higher values of $\lambda$ increase the importance of the adversary when minimizing the loss function of the tagger, which decorrelates the output from the tagger from the jet mass at the cost of worse classification when compared to standard neural networks.

In choosing a value of $\lambda$ to use in our analysis, we examined the different metrics and found that $\lambda = 50$ is where results start to saturate. Figure 16 shows the results of this parameter sweep using the three metrics used in the main body of this work. The ROC curves for the 2-, 3-, and 4-prong signals are shown in the top row. Darker shades correspond to lower values of $\lambda$. As expected, using lower values of $\lambda$ result in better classification. In the middle row, we have plotted the Bhattacharyya distance as a function of signal efficiency for every value of $\lambda$. The darkest curves look nearly identical to the Original NN results of Fig. 12, and sculpt the background the most, while the lightest curves (corresponding to higher values of $\lambda$) sculpt the least. From this row, we can see that the mass decorrelation as measured by the Bhattacharyya distance saturate at $\lambda = 50$. The bottom row shows a parametric plot of the Bhattacharyya distance and the background rejection, made by scanning across the signal efficiencies. For a fixed level of background rejection, we again see that the decorrelating benefits of the adversarial approach saturate at $\lambda = 50$.

# B  Comparison of histogram distances

We have used Bhattacharya distance in this work to quantify the sculpting of jet mass distribution from various jet tagging methods. This distance has the nice feature that it is normalized and therefore allows fair comparison across various methods. It is certainly not a unique choice. Another method used by ATLAS collaboration in Ref. [33] to quantify the mass distortion is the Jensen-Shannon distance, which is given as

$$d_{\text{JSD}}(P,Q) = \sqrt{\frac{d_{\text{KL}}(P, \frac{P+Q}{2}) + d_{\text{KL}}(Q, \frac{P+Q}{2})}{2}} \,, \tag{10}$$

where $d_{\text{KL}}$ is the Kullback-Leibler divergence, given by

$$d_{\text{KL}}(P,Q) = \sum_i p_i \log \frac{p_i}{q_i} \,, \tag{11}$$

$p_i, q_i$ being the value of the distribution $P,Q$ in bin $i$. For us, $P$ is the background mass distribution before the application of a given tagger, and $Q$ is the background mass distribution

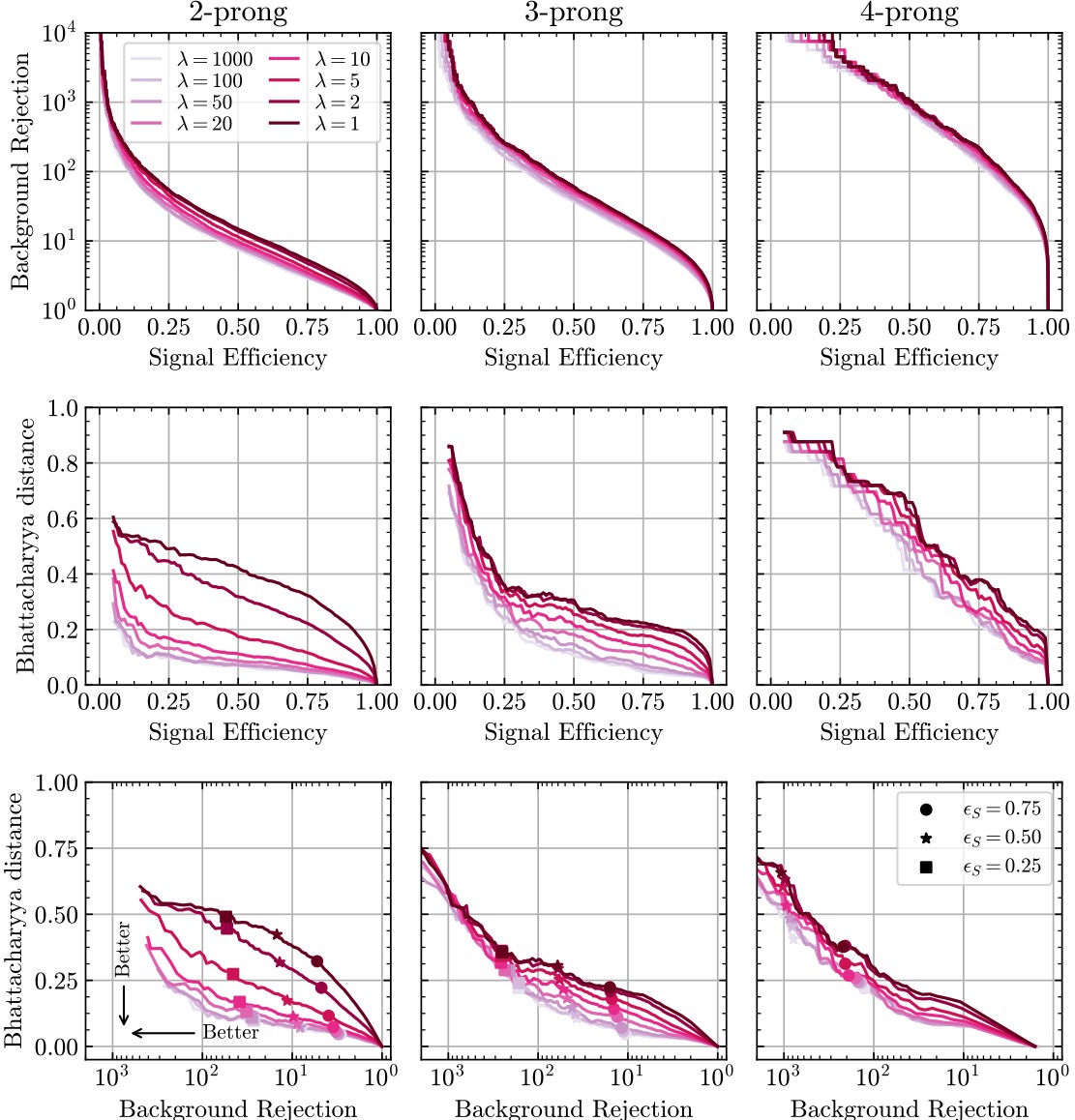

Figure 16: The top row shows the ROC curves for all of the adversarially-trained neural networks tasked with distinguishing the 2-, 3-, and 4-prong signal jets from the QCD background. Lighter shades correspond to increasingly larger values of $\lambda$. Larger values of $\lambda$ put an increased emphasis on making the network output less dependent on the mass, at the cost of worse classification. The middle row shows how the Bhattacharyya distance for the QCD background changes as tighter cuts are made on the network output. As expected, higher values of $\lambda$ lead to less sculpting than lower values of $\lambda$. The bottom row shows a parametric plot of the Bhattacharyya distance for the QCD background versus the background rejection. The adversarially-trained networks are all able to achieve similarly large background rejections, but networks using higher values of $\lambda$ are able to reject much of the background while preserving the profile of the underlying distribution. All three rows show that the benefits of adversarial training saturate at $\lambda = 50$.

after the application of a given tagger. In Fig. 17, we compare how the two distances $d_{\mathrm{B}}$ and $d_{\mathrm{JSD}}$ compare. We see that the two distances are very similar to each other for 2-pronged and

3-pronged signals, while have some differences in the 4-pronged case. The general shape is the same however, and one can be chosen over the other without biasing any inferences.

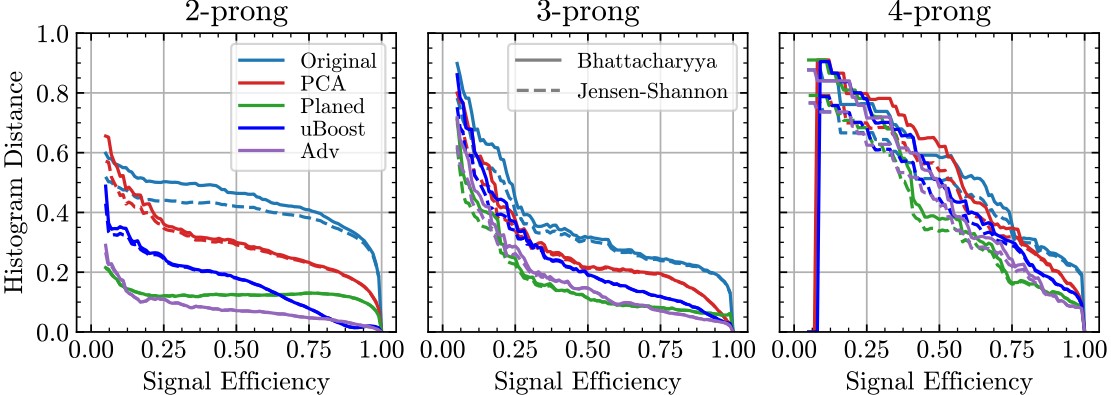

Figure 17: Comparison of Bhattacharyya distance and Jensen-Shannon distance for 2-, 3-, and 4-pronged signals, as a function of signal efficiency for various decorrelation methods studied in this work. The general trend for both metrics is seen to be the same.

## C    Histogram Sculpting Comparison

Here we show a qualitative comparison of all of the decorrelation methods for all of the different pronged signals considered in the main body of this work. The figures are organized as follows: the leftmost column shows the single-variable benchmark, $\tau_N^{(1)}/\tau_{N-1}^{(1)}$ ($N = 2, 3, 4$ for 2-/3-/4-pronged signal), as well as the Designed Decorrelated Tagger for the 2-prong signal; the middle column shows how the BDT benchmark sculpts the background, followed by all of the BDT based decorrelation methods studied in this work—uBoost, Planing, and PCA; the right column shows how the NN benchmark sculpts, followed by all of the NN based methods studied, namely Adversarial NNs, Planing, and PCA. A legend is provided in the lower left of each figure to remind the reader which colors correspond to which cuts on the signal efficiency, $\varepsilon_S$.

Figure 18 shows the comparison of methods for the 2-pronged signal, Fig. 19 shows this comparison for the 3-pronged signal, and Fig. 20 shows the comparison for the 4-prong signal.

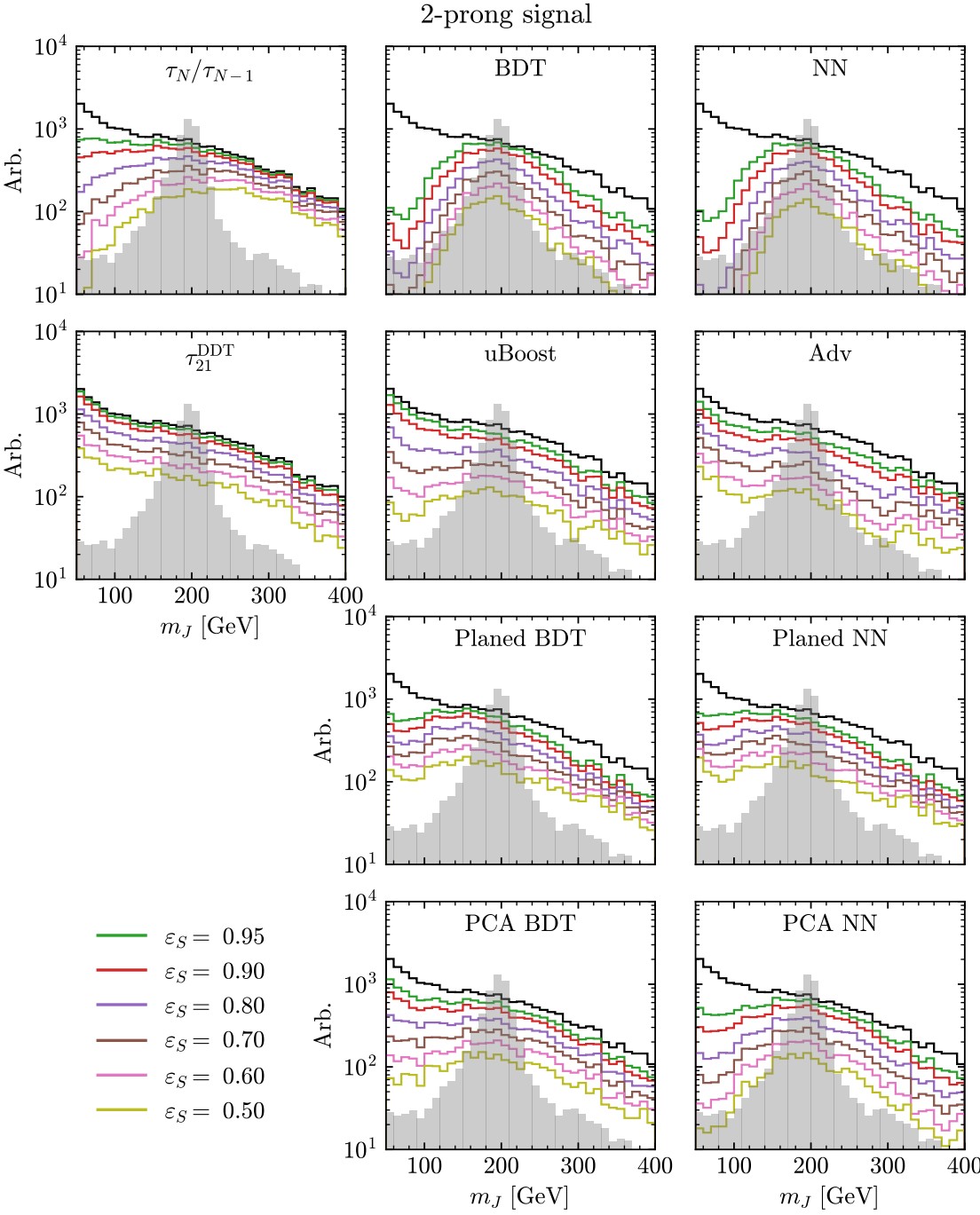

Figure 18: Comparison of all decorrelation methods to the benchmarks for the 2-prong signal. $\tau_N/\tau_{N-1}$ is $\tau_2/\tau_1$.

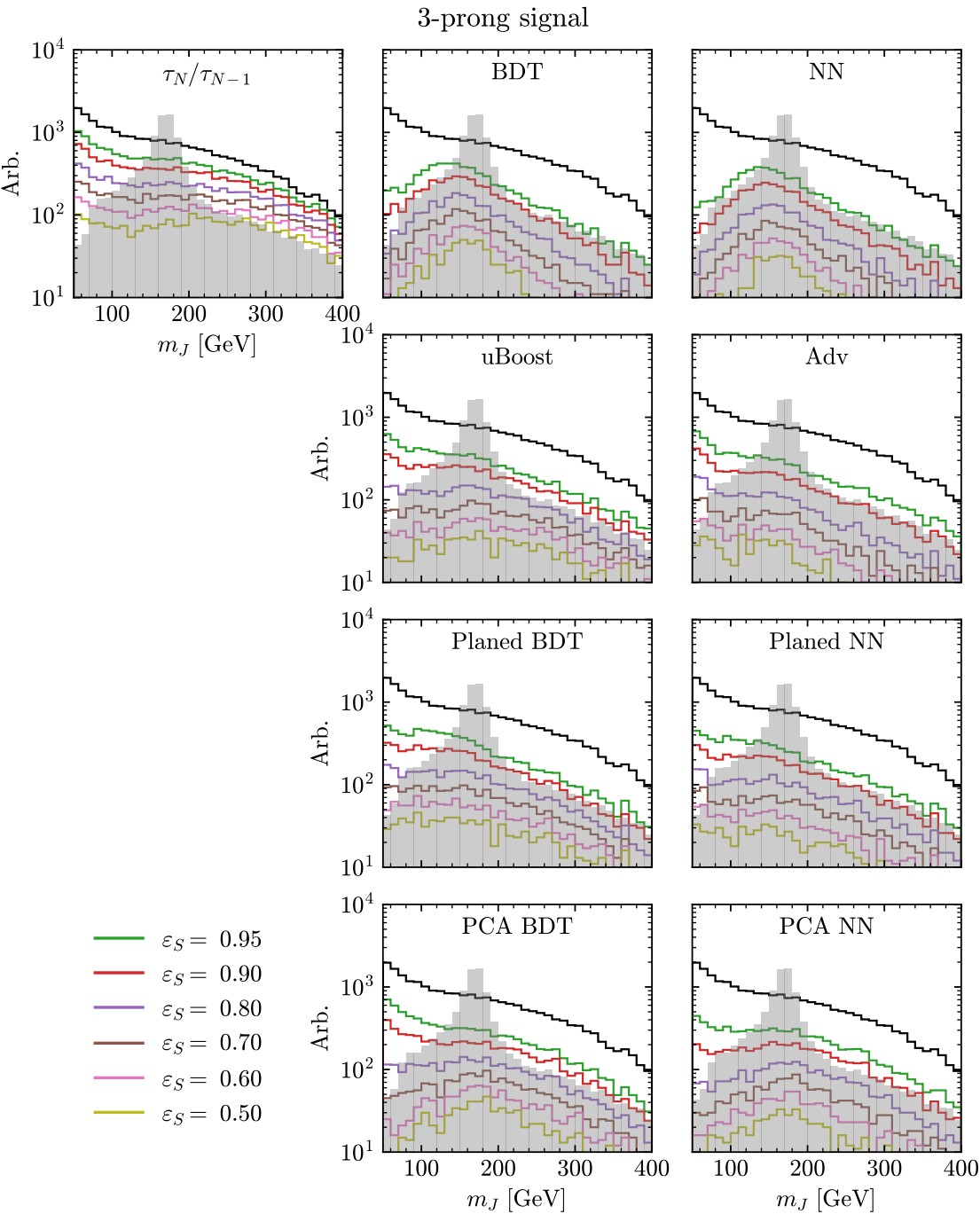

Figure 19: Comparison of all decorrelation methods to the benchmarks for the 3-prong signal. $\tau_N/\tau_{N-1}$ is $\tau_3/\tau_2$.

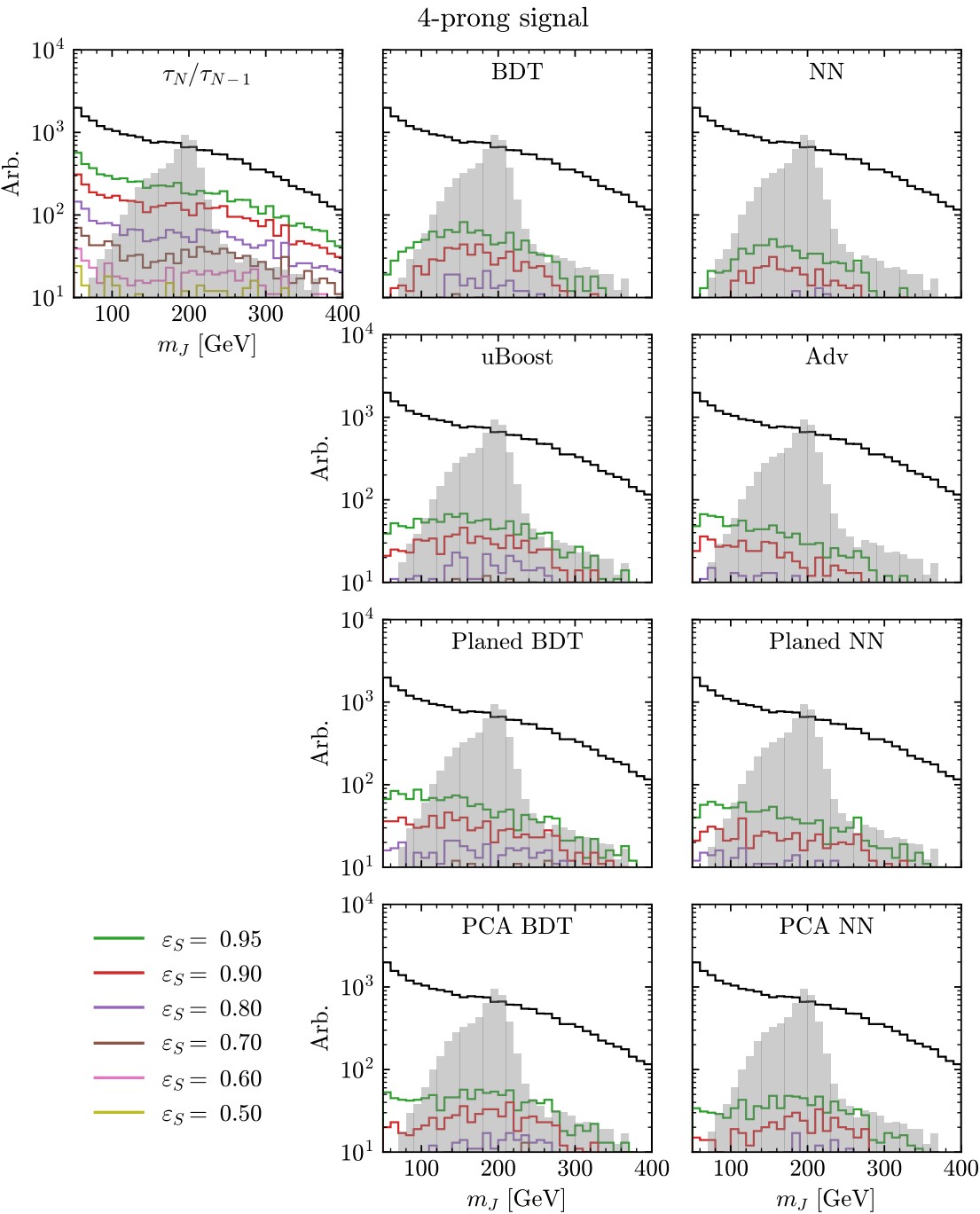

Figure 20: Comparison of all decorrelation methods to the benchmarks for the 4-prong signal. $\tau_N/\tau_{N-1}$ is $\tau_4/\tau_3$.

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
