# Peer review of "Mass Agnostic Jet Taggers"

_SciPost Physics, doi:SciPost Phys. 8, 011 (2020)_

## Round 1 · Referee Report · Anonymous · 2019-10-10

Strengths

(1) The article provides a useful benchmarking of different techniques that can be used to produce unbiased jet mass distributions for background events when using jet substructure techniques.

(2) Although the techniques used to decorrelate the jet mass information have all previously been presented separately in the literature, this article brings all the techniques into a single analysis and demonstrates that data augmentation techniques have similar performance to augmenting training methods.

(3) This is very relevant study at the current time, where jet substructure and multivariate analysis techniques are used in many searches for new physics beyond the SM.

Weaknesses

(1) The point of producing unbiased jet mass distributions is to help control systematics. However, systematics are not actually investigated in this article.

Report

This is a useful benchmarking of the different techniques that can be used to produce unbiased jet mass distributions for background events when using jet substructure techniques. The comparable performance of data augmentation to augmented training methods has implications for what methods are used at the LHC experiments.

Requested changes

(1) Background rejection definition: the analysis uses a wide mJ region (50<mJ<400GeV) and the background rejection is presumably defined over this range. However, it may be more useful to define the background rejection over a narrower range, such as a window that contains 95% of the signal. The reason is that the decorrelation approaches (augmented data or augmented training) result in a reduced background rejection ability, but it is not clear that the background rejection is much affected directly underneath the signal peak. Is there a good reason for using such a wide mJ range to define the background rejection? If not, then a narrower mJ range would also more clearly distinguish the background rejection ability from the background sculpting (which occurs across a much larger mJ range). Please consider making this change.

(2) The large Bhattacharayya distances for 4-prong events in Figs 12,14,15 give the impression that the mJ distributions are more sculpted than the 2-prong or 3-prong events. However, you point out in the text that the mJ distributions in Appendix C do not appear to support this impression. I agree and think that this is an illusion simply due to the different choices of x-axis range for 2-prong and 4-prong events. Over the range 10^3 - 10^0, the 2-prong, 3-prong and 4-prong show similar features and it is not clear that the 4-prong distributions are much more sculpted than the 2-prong. Ideally, it would be good to fix the binning on the plots to be the same so that this conclusion is easier to make.

More minor changes:

(3) section 2: more details on the exact generation of the QCD background is needed. Presumably this is 2->2 LO scattering.

(4) equation 1: the variable beta is not defined and this makes the subsequent understanding of the variable (and the ensemble of variables in eq 4) rather difficult.

(5) fig 4 caption and elsewhere. Please explain the coloured lines in more detail, i.e. the main text links each curve colour to explicit values of signal efficiency. This information should be added to the plot or caption.

(6) sec 4.1: I think more explanation is needed as to why unaltered input data is “better for calibration and other systematics”. Naively, I would think that the experimental/modelling systematics can easily be propagated through any MV technique.

---

## Round 1 · Referee Report · Tilman Plehn · 2019-10-18

Strengths

1- very nice collection of cutting-edge approaches
2- even nicer numerical comparison of these methods

Weaknesses

1- presentation not as clear as it could be
2- executive summary/bottom line missing (for all our experimentalist friends who do not really read papers)

Report

The paper is a really nice and timely study and will be extremely helpful for the future development of ML jet taggers. The only issue I have is that the authors discuss the individual approaches at great length and then end up with a relatively unstructured and hard-to-read discussion without a clear bottom line.

Requested changes

1- please shorten the abstract
2- separate cleanly between MV and modern ML methods somehow?
3- enlarge pictures, like Fig.4 etc
4- what us Tab.2 really meant to say? That NN is better but more expensive?
5- I find the discussion of all tools and all prong-nesses hard to follow and overwhelming. Maybe structure this a little more?
6- at the end it is not clear which approaches win and why.
7- conclusion is too long for senior people with little time.

---

## Round 1 · Referee Report · Anonymous · 2019-10-31

Strengths

1. excellent review of the potential and issues of fat-jet taggers
2. well structured systematic exploration of the mass-correlation issue, and the degree to which it is ameliorated by different techniques
3. important conclusion on relative performance of data/training augmentation methods for decorrelation, especially given the inflated training cost of the currently established methods

Weaknesses

1. lack of clarity on how analysers might optimise the trade-offs of training cost against the ROC characteristics
2. unclear whether data-augmentation has any negative effects in terms of CPU at application time, or the statistical power of the experimental datasets
3. verbose conclusions obscure the main take-away message

Report

This is a very nice and valuable paper: an important and systematic approach to optimising how we approach this increasingly important method. It is clear, well-presented, and wilth figures that make the point well. In addition it acts as a nice review of the area, from the issues of correlation to the state of the art in taggers and their decorrelation. The results will be a useful tool for those working in this area -- I just think that the conclusions can make the key points more clearly, and have suggested how below.

The issue of CPU cost is interesting, but for an experimental analysis the training cost of NNs is tiny compared to the overall data processing... it may not be the main driving issue for applications. However, the knowledge that planing and PCA give comparable performance more cheaply might mean that they are useful fast-turnaround prototyping tools for a final analysis based on a method that is more expensive to train. This leads to a second question: are the data-augmentation methods more expensive to _apply_, given that experiment data samples are very large, and are likely to be reprocessed many times during an analysis? And do they reduce the statistical power of the data (given that its statistics are precious, particularly on distribution tails, and reweighting reduced statistical power)? These issues will in general feed into the significance of a discovery or exclusion based on these methods, which will always be the primary factor in choosing a method: hopefully a future work will be able to look explicitly at a phenomenological proxy for the significance, factoring in systematics, likelihoods, and optimisation of the specific working tagger point.

p3, para 1: it's very nice that you consider this issue enough to introduce this systematics-enhanced S/sqrt(B + sigsys^2) measure... a pity that it couldn't be used as an unambiguous metric later in the paper: maybe in a future study.

p3, para 2: a bit strange to specifically refer to tau21 here, when that variable is only introduced later... but most readers will already know what it means, so up to you what you do with it!

p3, para 3: "are shown to be efficient in benchmark cases." -> begging for a reference to where this is shown

p4, para 2: you explain the origin of the 4-prong jets explicitly, but not the 2- and 3-prong cases. It takes a little inspection of the table and in particular the mKK variable to realise that in the 2P case the ZKK is light and hence itself boosted, while in the 3P and 4P cases the ZKK is unboosted and the multi-prong jet results from its fully hadronic decays. It would be better to make this explicit and save the reader some detective work. And perhaps also note that the "4P" process will also create 2P jets if one of the W/Z radion decay products decays leptonically (particularly a Z->nunu, in which case there is no hard lepton overlapping the jet, which would usually lead to deletion of the jet in overlap removal)

p4, para 3: this is good detail for reproducibility. Perhaps consider publishing the MadGraph and Delphes steering cards, and Delphes analysis code as arXiv ancillary files, to further assist "users" of your work.

p6, para 1: the AUC characterises the _overall_ performance of the classifier, for all working points considered equally. Maybe worth noting that for very differently shaped ROC curves it may not be the most relevant measure, in that a classifier with a lower AUC may have a more desirable specific working point if the two classifier ROCs cross. For these classifiers, as shown in Fig 4, the ROCs are so similar in shape and have no crossings that it looks a reasonable measure.

p6, para 3: it would be useful to clarify explicitly that this X basis is used by both multivariate classifiers: it is currently introduced but without connecting it to what follows

p8, para 3: expressing the cuts in terms of achieved efficiencies seems a strange way round: the cuts are tightened to increase background rejection, not to reduce signal efficiency! But of course they are monotonically related.

p10, para 1: the statement that these are similar tasks could do with a reference if possible

p10, para 2: can you clarify what variable the histogram is binned in? I am unclear about the use of m=m_i ... is the histogram in m or in X? Or both?!

p11, para 2: it's not clear here which sort of MV classifier you are using... but the word network appears a few times, hinting that you've now dropped the BDT: is this correct? I think the same comment applies to the PCA decorrelation benchmarking. EDIT: I see later that the BDT comes back, in the context of uBoost. Do the planing and PCA methods only apply to NNs? EDIT2: No, the BDTs reappear in Fig 13! So they really are just missing from this section and Fig 5, for no particular reason?

p12, para 3: is this AUC (4% drop) comparable to the 11% drop in Sec 3.2.2? Not clear, since the MV used in each isn't clear.

Fig 9: what is "Orig" in the ROC plot? Aren't there two "originals" here, BDT for uBoost and NN for Adv?

p17, para 2: I suppose it is for reasons of narrative, but a bit strange to treat the data-augmented methods first when establishing the benchmarks, then switch the order to augmented-training when analysing the degree of decorrelation achieved...

p18, para 1: it's not that QCD doesn't have prongs, it's that it most typically has a single _prong_ corresponding to the initiating parton. If it had no dominant axes at all, large-Nprong subjettinesses would fit QCD increasingly well!

p24, para 1: a legion of experimentalists may spit out their coffee over the use of the word "quite" in the first sentence!

p24-25: I would just comment that this conclusion is very verbose, and that obscures the fact that you actually have an important result that can be expressed pithily, and will be more impactful if you do so. I won't insist, but only suggest that you reduce its length by cutting down some of the repetition in the first half. The second half contains a lot of new reflection, comparison and next-ideas material which IMHO would fit best in an Outlook/Prospects section before a short conclusion.

p25: In the interests of reproduction, putting the code for reproduction also in the arXiv ancillary or on Zenodo would be ideal -- github is likely to be around for a long time, but probably not forever.

Appendices: these seem appropriate, and it's nice to have the extra detail.

Requested changes

1. clarify the origins of each prong multiplicity in the simulation: currently only explicit for 4P
2. add or explain the apparent absence of BDT performance in Sec 3.2
3. explain whether there are any penalties in terms of application-time CPU (e.g. on experimental data) from the data-augmentation decorrelation methods
4. explain whether data-augmentation decorrelation has any impact on the statistical power of the experimental data
5. recommend to distinguish outlook from conclusions, to make the latter clearer

---

## Round 5 · Referee Report · Tilman Plehn (Referee 2) · 2019-12-17

Report

Thank you for taking into account my comments, very nice work!

---

## Round 5 · Referee Report · Anonymous (Referee 3) · 2019-12-17

Report

Thanks to the authors for taking on board most of the points I made, and I don't regard the remainder as problems. I do think the statistical impact of reweighting will need to be considered in practice -- there is increasing awareness that MC samples are also expensive, and a danger for future experimental runs that their statistics even without reweighting will be a limiting factor. But this is an additional point of context, and on the raw science of comparing decorrelation methods this paper is a valuable contribution.

---

## Round 5 · Author Response

We thank the referees for their time and thoughtful comments. The draft is greatly improved, thanks to them.

---

## Round 5 · List of Changes

-We shortened the abstract as Referee 2 suggested.

-As per the comments made by Referee 1 and 3 regarding the details of how the signal and background events were generated, we added details to Sec. 2 about how the 2- and 3-prong signals were generated, and clarify that the background events were generated at leading order in the QCD coupling.

-Regarding Referee 2’s concern regarding the distinction between multivariate and machine learning methods, we acknowledge the distinction between the two and make clear that we are using the two terms interchangeably in the discussion that follows. This change is made at the beginning of Sec. 3

-Referee 1 asked that we define the variable beta in Eq. 1, and we have clarified that beta is a real number in Sec. 3.1. To remove any possible confusion with the variable beta used in the uBoost algorithm, we have renamed that hyperparameter as beta_{u} to make clear that this is a different beta than the one used in defining the N-subjettiness variables.

-Referee 3 asked that we clarify that the X basis introduced in Eq. 4 is used by both multivariate classifiers, and we have added a sentence immediately following Eq. 4 making this clarification.

-Referee 1 asked that we explain the colored lines appearing in Fig. 4 and elsewhere. In the captions of Figs. 4,5, and 9, we explain that the colored lines correspond to different values of the signal efficiency.

-Referee 3 requested that we add or explain the absence of Boosted Decision Tree performance in Sec. 3.2, as well as explaining whether there are any penalties in terms of application time for the data augmentation techniques. In Sec. 3.2, prior to explaining any of the data augmentation techniques, we explain that we are only showing results for the Neural Networks when introducing these techniques, but that they can be applied to either. We also explain that these methods are fast, and that we expect the application-time cost to be minimal.

-Referee 1 had concerns regarding our definition of the background rejection over the whole mass range we consider. To the beginning of Sec. 4, we addressed these concerns by explaining how we expect our this choice to give qualitatively the same results that we would have found if we defined the background rejection over a narrower window centered around the signal, since all of the taggers considered in this work are tasked with keeping the background rejection constant over the whole mass range.

-Referee 2 asked us to explain what the takeaway from Table 2 is meant to be. We clarified this in the caption.

-Referee 3 pointed out that our statement regarding the prongedness of the QCD background in Sec. 4.1 was imprecise. We have made this correction.

-Referee 1 was concerned that the different choices for the x-axes in Figs. 12,14, and 15 incorrectly give the impression that the 4-prong events are sculpted more than the 2- or 3-prong signals. We changed the x-axes of these figures so that the background rejection for all of the signals are in the range 10^3-10^0. The trend is much easier to see now.

-Referee 2 and 3 asked that we distinguish our outlook from the conclusion to make the latter both shorter and easier to understand. We added a new section, Sec. 5, where we discuss the outlook and future work. We also shortened our conclusion---now in Sec. 6---to make the main takeaways of our work more clear.

-Regarding Referee 3’s concerns about whether the data augmentation techniques have any impact on the statistical power of the experimental data, we explain at the beginning of Sec. 5 that we don’t expect this to be the case since the PCA based approach involves only a linear transformation of the data, and Planing only requires that the training data (likely from Monte Carlo) be reweighted.

---

## Editorial Decision

published